# Eco1-dependent cohesin acetylation anchors chromatin loops and cohesion to define functional meiotic chromosome domains

**Rachael E Barton†, Lucia F Massari†, Daniel Robertson, Adèle L Marston\***

The Wellcome Centre for Cell Biology, Institute of Cell Biology, School of Biological Sciences, Michael Swann Building, Max Born Crescent, Edinburgh, United Kingdom

**Abstract** Cohesin organizes the genome by forming intra-chromosomal loops and inter-sister chromatid linkages. During gamete formation by meiosis, chromosomes are reshaped to support crossover recombination and two consecutive rounds of chromosome segregation. Here, we show that meiotic chromosomes are organized into functional domains by Eco1 acetyltransferase-dependent positioning of both chromatin loops and sister chromatid cohesion in budding yeast. Eco1 acetylates the Smc3 cohesin subunit in meiotic S phase to establish chromatin boundaries, independently of DNA replication. Boundary formation by Eco1 is critical for prophase exit and for the maintenance of cohesion until meiosis II, but is independent of the ability of Eco1 to antagonize the cohesin release factor, Wpl1. Conversely, prevention of cohesin release by Wpl1 is essential for centromeric cohesion, kinetochore mono-orientation, and co-segregation of sister chromatids in meiosis I. Our findings establish Eco1 as a key determinant of chromatin boundaries and cohesion positioning, revealing how local chromosome structuring directs genome transmission into gametes.

**\*For correspondence:**
adele.marston@ed.ac.uk

†These authors contributed equally to this work

## Editor's evaluation

With a large quantity and high quality of data set, this paper elegantly shows the role of Eco1-dependent Smc3 acetylation plays a role in the establishment of chromatin boundary during yeast meiosis, which is independent of meiotic DNA replication. Eco1-directed boundary formation, which is not counteracted by Wapl, is critical for prophase I exit and sister chromatid separation in meiosis II. In addition, Eco1 antagonizes Wapl-mediated cohesin removal at centromeres for proper sister cohesion and mono-orientation of sister kinetochores in meiosis I.

## Introduction

The cohesin complex defines genome architecture to support DNA repair, gene expression, and chromosome segregation (*Davidson and Peters, 2021*). Core cohesin is a DNA translocase comprising a V-shaped heterodimer of two structural maintenance of chromosomes proteins, Smc1 and Smc3, whose two ATPase heads are connected by a kleisin subunit. Cohesin folds the genome through ATP-dependent extrusion of intra-molecular DNA loops. Cohesin also entraps newly replicated sister chromatids within its tripartite structure to establish the cohesion needed for chromosome segregation. Loop extrusion and cohesion are biochemically distinct and dependent on cohesin accessory proteins, although the mechanisms are not completely understood (*Srinivasan et al., 2018*). Loading of cohesin onto DNA requires the Scc2-Scc4 (NIPBL-MAU2 in mammals) complex, which also drives loop extrusion (*Ciosk et al., 2000*; *Davidson et al., 2019*; *Petela et al., 2018*; *Srinivasan et al.,*

*2019*). Chromosomal cohesin is destabilized by the cohesin release factor, Wpl1/Rad61 (WAPL in mammals), an activity that is counteracted by acetylation of cohesin's Smc3 subunit by the Eco1 acetyltransferase (*Rolef Ben-Shahar et al., 2008*; *Unal et al., 2008*).

Cohesin-mediated loops are positioned by boundary elements. In mammalian interphase cells, the insulation protein CTCF anchors cohesin at the sites of long-range interactions (*Haarhuis et al., 2017*; *Rao et al., 2017*; *Schwarzer et al., 2017*; *Wutz et al., 2017*). In yeast, CTCF is absent, but chromosomes are nevertheless organized into cohesin-dependent loops in mitosis and convergent genes, which are known to accumulate cohesin, are found at loop boundaries (*Costantino et al., 2020*; *Lazar-Stefanita et al., 2017*; *Lengronne et al., 2004*; *Paldi et al., 2020*; *Schalbetter et al., 2017*). Budding yeast pericentromeres provide an exemplary model of how a functional chromosome domain is folded. Cohesin loaded at centromeres extrudes loops on both sides until it is stalled by convergent genes at flanking pericentromere boundaries, thereby establishing a structure that facilitates chromosome segregation in mitosis (*Paldi et al., 2020*). Loop size and position are also controlled by cohesin regulators. Wpl1 restricts loop size, but does not affect their positioning (*Costantino et al., 2020*; *Dauban et al., 2020*; *Haarhuis et al., 2017*). Eco1 also limits long-range interactions, but additionally affects loop positioning (*Dauban et al., 2020*). Whether this is a direct effect, through Eco1-mediated acetylation and inhibition of loop-extruding cohesin, or indirect, as a result of acetylated cohesive cohesin forming a barrier to translocating cohesin, is unclear. Cells lacking Eco1 are inviable due to cohesion defects, which can be rescued by removal of Wpl1 to restore viability (*Rolef Ben-Shahar et al., 2008*; *Rowland et al., 2009*; *Sutani et al., 2009*; *Unal et al., 2008*). However, cells lacking both Wpl1 and Eco1 show an additive increase in long-range interactions (*Dauban et al., 2020*). Together, these observations suggest that loop positioning is not essential for viability. Consistent with its essential function in establishing cohesion, in mitotically growing cells, acetylation of Smc3 occurs in S phase and is largely dependent on DNA replication (*Beckouët et al., 2010*; *Rolef Ben-Shahar et al., 2008*). As cells exit vegetative S phase, Eco1 is degraded, which prevents deposition of Smc3 acetylation post-S phase (*Lyons and Morgan, 2011*). Overexpression of Eco1 in metaphase-arrested cells is sufficient to acetylate newly loaded cohesin, implying that Eco1 might also be capable of acetylating Smc3 independently of DNA replication (*Beckouët et al., 2010*; *Lyons and Morgan, 2011*). Moreover, in the presence of DNA damage, Eco1 is stabilized after S phase and acts to acetylate cohesin on replicated chromosomes, albeit on the Scc1, rather than Smc3, subunit (*Heidinger-Pauli et al., 2009*; *Lyons et al., 2013*). In mammals, ESCO2 similarly acetylates SMC3 during S phase to establish cohesion (*Alomer et al., 2017*) while, during interphase, an additional family member, ESCO1, is active and contributes to boundary formation in chromatin looping (*Wutz et al., 2020*). Together, these observations indicate that Eco1-dependent cohesin acetylation can both position loops and maintain cohesion.

During gamete formation by meiosis, chromosomes undergo extensive restructuring, underpinned by cohesin-dependent chromosome looping and cohesion. In many organisms, a meiosis-specific kleisin, Rec8, enables functions that cannot be carried out by canonical kleisin, Scc1/Rad21 (*Severson et al., 2009*; *Tachibana-Konwalski et al., 2010*; *Tóth et al., 2000*; *Yokobayashi et al., 2003*). During meiotic prophase, chromosomes comprise a dense array of chromatin loops emanating from cohesin-rich axes that are zipped together in homologous pairs by a central core – the synaptonemal complex (*Cahoon and Hawley, 2016*). Budding yeast Rec8-cohesin anchors loops at their base (*Muller et al., 2018*; *Schalbetter et al., 2019*) and supports crossover recombination and DNA repair to allow prophase exit (*Klein et al., 1999*). Between prophase I and metaphase I, Wpl1 removes a fraction of chromosomal Rec8 (*Challa et al., 2016*; *Challa et al., 2019*). WAPL also promotes meiotic release of cohesin in *Arabidopsis thaliana* and *Caenorhabditis elegans*, and reduces loop number in mouse oocytes, although the kleisin target differs between organisms (*Crawley et al., 2016*; *De et al., 2014*; *Silva et al., 2020*). Following prophase exit, two distinct meiotic divisions ensue (reviewed in *Duro and Marston, 2015*). During meiosis I, sister kinetochores are mono-oriented to ensure sister chromatid co-segregation. In budding yeast, the monopolin complex fuses sister kinetochores, while in fission yeast and mammals centromeric Rec8-cohesin directs sister chromatid co-segregation (*Chelysheva et al., 2005*; *Monje-Casas et al., 2007*; *Parra et al., 2004*; *Sakuno et al., 2009*; *Sarangapani et al., 2014*; *Severson et al., 2009*). Homolog segregation at meiosis I is triggered by separase-dependent cleavage of Rec8 on chromosome arms, while pericentromeric cohesin is retained and cleaved only at meiosis II to allow sister chromatid segregation. How cohesin-dependent loop formation and cohesion

are spatially and temporally regulated to establish distinct functional chromosome domains in meiosis remains unclear.

Here, we identify Eco1 acetyltransferase as a key determinant of localized meiotic chromosome structure in budding yeast. In meiosis, Eco1 acetylates Smc3 both linked to and independently of DNA replication and is critical for viability, even in the absence of Wpl1. Eco1 counteracts Wpl1 to allow centromeric cohesion establishment and thereby kinetochore mono-orientation. In contrast, arm cohesion and prophase exit require an Eco1 function other than Wpl1 antagonism. We show that anchoring of chromatin loops is the critical Wpl1-independent function of Eco1. While Eco1 and Wpl1 independently restrict chromatin loop size in prophase I, only Eco1 is critical for boundary formation, notably at pericentromere borders. We propose that cohesin acetylation by Eco1 traps both loop extruding and cohesive cohesin complexes at boundaries to define a chromosome architecture that is essential for meiotic recombination and chromosome segregation.

## Results

### Eco1 acetylates cohesin during meiotic S phase, independently of DNA replication

To determine the timing of Eco1-dependent Smc3 acetylation during meiosis, wild-type cells carrying functional *ECO1-6HIS-3FLAG* (*Figure 1—figure supplement 1A*) were arrested in G1 prior to meiotic S phase and then released (by conditional *IME1/IME4* expression; *Berchowitz et al., 2013*), allowing synchronous meiotic DNA replication and nuclear divisions (*Figure 1A and B*). Eco1 levels increased after meiotic entry, were maximal around the time of meiotic S phase, and declined after DNA replication, while Smc3-K112,K113 acetylation (henceforth Smc3-Ac), detected using a verified antibody (*Figure 1—figure supplement 1B and C*), accumulated after Eco1 appearance, persisted throughout the meiotic divisions, and declined during meiosis II (*Figure 1C*).

To test whether DNA replication is required for Smc3 acetylation in meiosis, we analyzed *pSCC1-CDC6* (*cdc6* meiotic depletion; *cdc6-md*) and *clb5Δ clb6Δ* cells that fail to assemble or fire pre-replicative complexes, respectively, resulting in little or no replication in meiotic S phase (*Figure 1D*; *Brar et al., 2009*; *Hochwagen et al., 2005*; *Stuart and Wittenberg, 1998*). In *cdc6-md* and *clb5Δ clb6Δ* meiotic cells, Smc3-Ac appeared with comparable timing to wild type and was only modestly reduced (*Figure 1E and F*). In contrast, Smc3-Ac was greatly diminished in cells lacking the meiotic cohesin kleisin subunit (*rec8Δ*) (*Figure 1E and F*). Because DNA double-strand breaks trigger Eco1-dependent cohesion establishment in mitotic cells (*Ström et al., 2007*; *Unal et al., 2007*), we tested whether Spo11 endonuclease-induced meiotic double-strand breaks were required for Smc3-Ac. However, preventing double-strand break formation (*spo11Δ*) did not reduce Smc3-Ac levels whether chromosomes were replicated or not (*Figure 1—figure supplement 2*). We conclude that cohesin acetylation on its Smc3 subunit during meiotic S phase occurs independently of DNA replication and programmed double-strand break formation.

### Eco1 is essential for meiosis, even in the absence of *WPL1*

Eco1 is essential for vegetative growth and *eco1Δ* viability is restored by deletion of *WPL1* (*eco1Δ wpl1Δ* cells) (*Rolef Ben-Shahar et al., 2008*). To examine Eco1 function specifically in meiosis, we fused it to the FKBP-rapamycin-binding domain (FRB) to allow rapamycin-induced anchoring out of the nucleus (*Haruki et al., 2008*; *Figure 1G*; see also Materials and methods). Growth of *ECO1-FRB-GFP* cells was inhibited in the presence of rapamycin, but restored by deletion of *WPL1* (*Figure 1H*), consistent with successful anchor-away (*Rolef Ben-Shahar et al., 2008*; *Sutani et al., 2009*; *Unal et al., 2008*). Upon induction of meiosis in the presence of rapamycin, Smc3-Ac was undetectable in *ECO1-FRB-GFP* cells and full-length Rec8 persisted long after its expected time of degradation (*Figure 1I*). Furthermore, only a minor fraction of *ECO1-FRB-GFP* cells completed the first meiotic division (*Figure 1J*). Even in the absence of rapamycin, *ECO1-FRB-GFP* cells showed defects in meiotic progression, Rec8 degradation, and Smc3-Ac (*Figure 1—figure supplement 3*), despite supporting vegetative growth (*Figure 1H*). Therefore, the FRB-GFP tag on Eco1 specifically affects meiosis, and we employed the *ECO1-FRB-GFP* allele as a constitutive meiotic loss-of-function mutant. Henceforth, *ECO1-FRB-GFP* cells that were induced to sporulate in the presence of rapamycin are denoted as

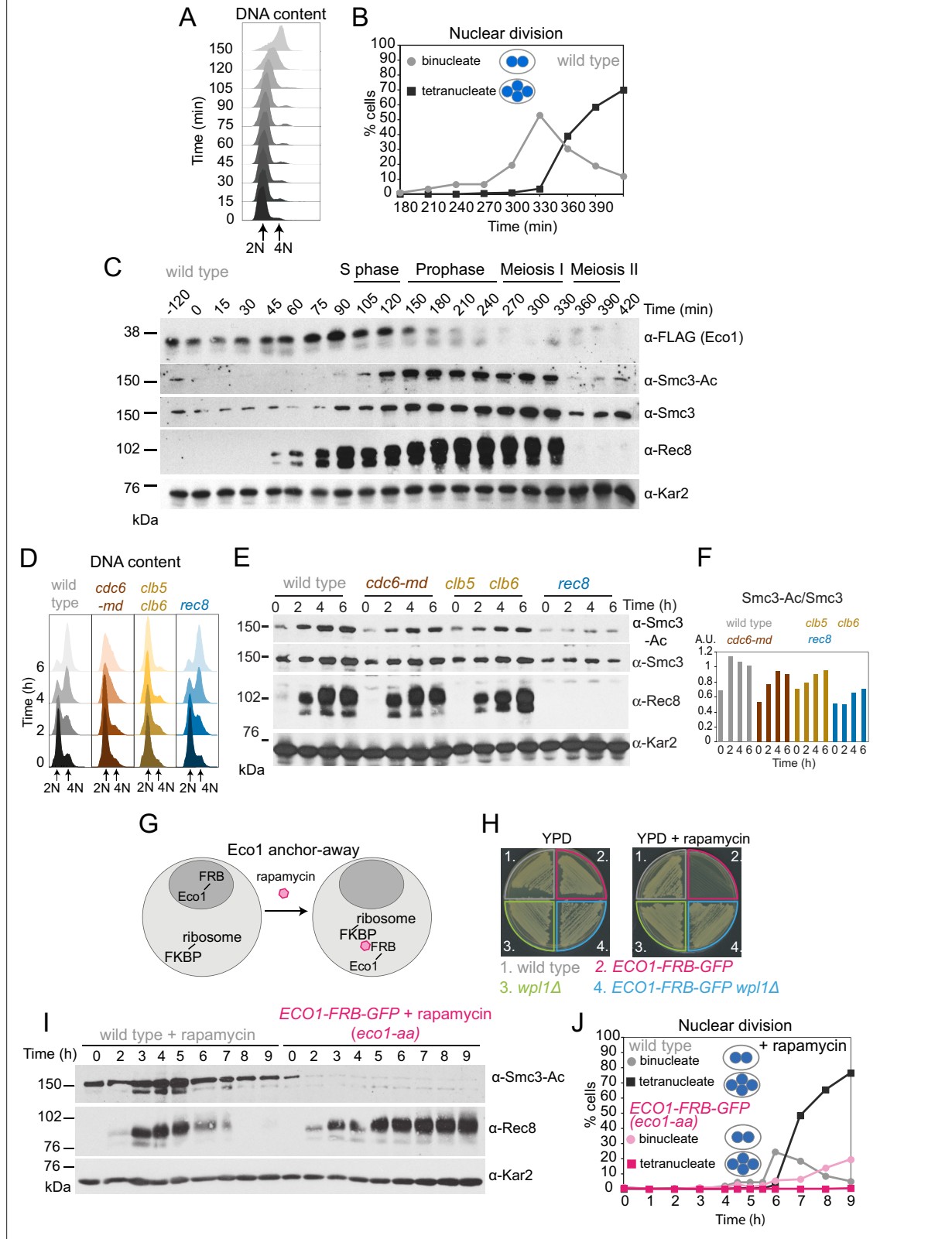

**Figure 1.** Eco1-dependent acetylation of Smc3-K112,113 occurs in meiotic S phase, independently of DNA replication. (**A–C**) Smc3-Ac is deposited in S phase, following Eco1 production. Wild type (strain AM21574) carrying *ECO1-6HIS-3FLAG* and *pCUP1-IME1 pCUP1-IME4* was released from a pre-meiotic S phase block 120 min after sporulation induction by addition of 25 µM CuSO₄. (**A**) S phase completion (4N) was monitored by flow cytometry. (**B**) The percentages of bi- and tetranucleate cells were scored at the indicated timepoints to monitor meiosis I and II nuclear division, respectively (n =

*Figure 1 continued on next page*

*Figure 1 continued*

200 cells per timepoint). (**C**) Western immunoblot shows the total cellular levels of Eco1-6HIS-3FLAG (α-FLAG), Smc3-Ac (α-Smc3-K112,113-Ac), Smc3 (α-Smc3), and Rec8 (α-Rec8) with Kar2 as a loading control (α-Kar2). (**D, E**) Bulk DNA replication is not essential for Smc3-Ac. Wild-type (AM11633), *cdc6-md* (AM28842), *clb5Δ clb6Δ* (AM28841), and *rec8Δ* (AM28843) cells carrying *ndt80Δ* were induced to sporulate and allowed to arrest in prophase I. (**D**) Flow cytometry shows DNA content. (**E**) Western immunoblot shows total cellular levels of Smc3-Ac (α-Smc3-K112,113-Ac), Smc3 (α-Smc3), Rec8 (α-Rec8), and Kar2 loading control (α-Kar2). (**F**) Quantification of Smc3-Ac normalized to Smc3 protein levels (A.U. = arbitrary units). (**G**) Schematic of the anchor-away system used to deplete Eco1 from the nucleus (*eco1-aa*). (**H**) The lethality of Eco1 anchor-away is rescued by deletion of *WPL1*. Haploid wild-type (AM13762), *ECO1-FRB-GFP* (AM22004), *wpl1Δ* (AM22440), and *ECO1-FRB-GFP wpl1Δ* (AM22981) strains of the anchor-away background (*RPL13A-FKBP12, fpr1Δ, tor1-1*) were plated on YPD or YPD + 1 μM rapamycin. (**I, J**) Eco1 is essential for meiotic progression. Anchoring-away Eco1-FRB-GFP reduces acetylation of Smc3-K112,K113, impairs cleavage of Rec8, and reduces nuclear divisions. Anchor-away wild-type (AM25532) and *ECO1-FRB-GFP* (AM22034) cells were induced to sporulate in the presence of 1 μM rapamycin. (**I**) Western immunoblot of whole-cell extracts showing Smc3-Ac (α-Smc3-K112,K113-Ac), Rec8 (α-Rec8), and Kar2 loading control (α-Kar2). (**J**) The percentages of bi- and tetranucleate cells were scored after DAPI staining at the indicated timepoints (n = 200 cells/timepoint).

The online version of this article includes the following source data and figure supplement(s) for figure 1:

**Source data 1.** Source images for the blot in *Figure 1C*.

**Source data 2.** Source images for the blot in *Figure 1C*.

**Source data 3.** Source images for the blot in *Figure 1C*.

**Source data 4.** Source images for the blot in *Figure 1C*.

**Source data 5.** Source images for the blot in *Figure 1C*.

**Source data 6.** Source images for the blot in *Figure 1E*.

**Source data 7.** Source images for the blot in *Figure 1E*.

**Source data 8.** Source images for the blot in *Figure 1E*.

**Source data 9.** Source images for the blot in *Figure 1E*.

**Source data 10.** Source images for the blot in *Figure 1I*.

**Source data 11.** Source images for the blot in *Figure 1I*.

**Source data 12.** Source images for the blot in *Figure 1I*.

**Figure supplement 1.** Tools to analyze Eco1-dependent Smc3 acetylation in meiosis.

**Figure supplement 1—source data 1.** Source images for the blot in *Figure 1—figure supplement 1C*.

**Figure supplement 1—source data 2.** Source images for the blot in *Figure 1—figure supplement 1C*.

**Figure supplement 1—source data 3.** Source images for the blot in *Figure 1—figure supplement 1C*.

**Figure supplement 2.** Smc3-Ac does not require meiotic recombination.

**Figure supplement 2—source data 1.** Source images for the blot in *Figure 1—figure supplement 1B*.

**Figure supplement 2—source data 2.** Source images for the blot in *Figure 1—figure supplement 1B*.

**Figure supplement 2—source data 3.** Source images for the blot in *Figure 1—figure supplement 1B*.

**Figure supplement 2—source data 4.** Source images for the blot in *Figure 1—figure supplement 1B*.

**Figure supplement 3.** Eco1-FRB-GFP is nonfunctional in meiosis even in the absence of rapamycin.

**Figure supplement 3—source data 1.** Source images for the blot in *Figure 1—figure supplement 1A*.

**Figure supplement 3—source data 2.** Source images for the blot in *Figure 1—figure supplement 1A*.

**Figure supplement 3—source data 3.** Source images for the blot in *Figure 1—figure supplement 1A*.

*eco1-aa* (Eco1 *anchor-away*). We conclude that Eco1 acetylates Smc3 on residues K112,113 during S phase of meiosis, and that Eco1 is required for efficient meiotic division.

We reasoned that Eco1 activity early in meiosis may be required to counter Wpl1-dependent cohesin destabilization during any of the subsequent stages of meiosis. If this is the case, deletion of Wpl1 may overcome the impaired ability of *eco1-aa* cells to undergo the meiotic divisions (*Figure 1J*). Wpl1 promotes the non-proteolytic removal of cohesin between meiotic prophase and metaphase I (*Challa et al., 2016*; *Challa et al., 2019*). Prior to meiotic S phase, functional Wpl1-6HA (*Figure 2—figure supplement 1A*) gains an activating phosphorylation (*Challa et al., 2019*) and its overall levels increase, as visualized by Western blotting (*Figure 2A*) in cells synchronously undergoing S phase (*Figure 2B*) and the meiotic divisions (*Figure 2C*). Following S phase, and reminiscent of the sequential loss of cohesin from chromosome arms and pericentromeres, Wpl1 undergoes stepwise

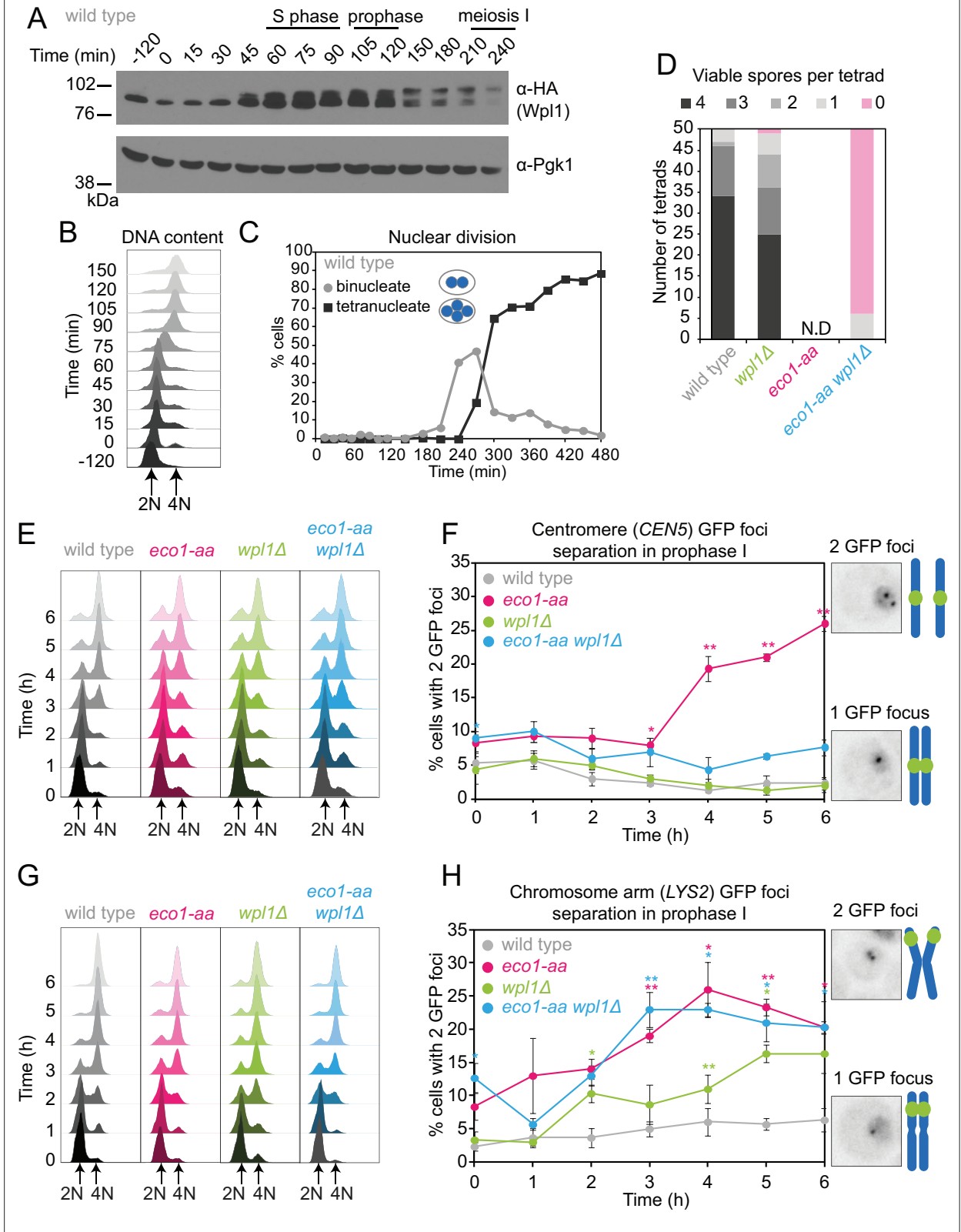

**Figure 2.** Counteracting Wpl1 is not the only essential role of Eco1 in meiosis. (**A–C**) Wpl1 is most abundant during meiotic S phase and prophase. Wild type (AM20916) carrying *WPL1-6HA* and *pCUP1-IME1 pCUP1-IME4* was induced to undergo synchronous meiotic S phase as described in *Figure 1A*. (**A**) Western immunoblot shows total protein levels of Wpl1-6HA (α-HA) with Pgk1 loading control (α-Pgk1). (**B**) Flow cytometry profiles and (**C**) nuclear division (n = 200 cells per timepoint) show the timing of bulk DNA replication and chromosome segregation, respectively. (**D**) Eco1 is essential for

*Figure 2 continued on next page*

*Figure 2 continued*

meiosis, even in the absence of Wpl1. Spore viability of wild-type (AM24170), *wpl1Δ* (AM24265), *eco1-aa* (AM24171), and *eco1-aa wpl1Δ* (AM24289) strains, sporulated in the presence of 1 μM rapamycin. N.D. = not done since it was not possible to find 50 tetrads in *eco1-aa* cells to measure spore viability. (**E, F**) Establishment of centromeric cohesion requires Eco1-dependent antagonism of Wpl1. Wild-type (AM27183), *wpl1Δ* (AM27186), *eco1-aa* (AM27185), and *eco1-aa wpl1Δ* (AM27184) anchor-away strains carrying heterozygous *CEN5*-GFP and *ndt80Δ* were induced to sporulate in 1 μM rapamycin, and the percentage of cells with two visible GFP foci was scored at the indicated timepoints (**F**). Meiotic progression was monitored as DNA content (**E**). (**G, H**) Chromosomal arm cohesion requires Eco1, even in the absence of Wpl1. Wild-type (AM27253), *wpl1Δ* (AM27256), *eco1-aa* (AM27255), and *eco1-aa wpl1Δ* (AM27254) anchor-away strains carrying heterozygous *LYS2*-GFP and *ndt80Δ* were treated and analyzed as described in (**E**) and (**F**). In (**F, H**), an average of three biological replicates is shown; 100 cells were scored for each timepoint in each experiment. Error bars show standard error; \*p<0.05, \*\*p<0.01, paired Student's *t*-test when compared to wild type.

The online version of this article includes the following source data and figure supplement(s) for figure 2:

**Source data 1.** Source images for the blot in *Figure 2A*.

**Source data 2.** Source images for the blot in *Figure 2A*.

**Figure supplement 1.** Wpl1-6HA is functional and expressed in meiosis.

**Figure supplement 1—source data 1.** Source images for the blot in *Figure 2—figure supplement 2B*.

**Figure supplement 1—source data 2.** Source images for the blot in *Figure 2—figure supplement 2B*.

**Figure supplement 2.** Eco1 is required for meiosis even in the absence of Wpl1.

degradation during meiosis I and II (*Figure 2—figure supplement 1B and C*). To test the idea that Eco1 allows meiotic progression by countering Wpl1, we sought to measure spore formation and viability (*Figure 2D*, *Figure 2—figure supplement 2*). However, deletion of *WPL1* in the *eco1-aa* background only slightly increased the formation, but not the viability, of spores, while *wpl1Δ* cells showed a small decrease in spore viability, as reported (*Challa et al., 2016*; *Figure 2D*, *Figure 2—figure supplement 2A*). Consistently, *WPL1* deletion did not restore nuclei division to *eco1-aa* cells (*Figure 2—figure supplement 2B*). Therefore, in contrast to vegetative cells, Eco1 is essential for meiosis, even in the absence of Wpl1. Sporulating *ECO1-FRB-GFP* cells in the absence of rapamycin increased sporulation efficiency, but only slightly improved viability (*Figure 2—figure supplement 2C*).

## Distinct requirements for Eco1 in cohesion establishment at centromeres and on chromosome arms

To determine whether *eco1-aa* cells establish functional cohesion during S phase, we labeled one homolog either at a centromere (*CEN5*-GFP) or at a chromosomal arm site (*LYS2*-GFP) and scored the percentage of cells with two GFP foci (indicating defective cohesion) as cells progressed through meiotic S phase into a prophase I arrest. In wild-type prophase cells, sister chromatids are tightly cohered and a single focus is visible. In contrast, *eco1-aa* cells showed a profound cohesion defect (*Figure 2E–H*). Remarkably, we observed distinct effects of *WPL1* deletion in *eco1-aa* cells on centromeres and chromosome arms: deletion of *WPL1* restored cohesion at the centromere (*CEN5*-GFP), but not at the chromosomal arm site (*LYS2*-GFP) (*Figure 2F and H*). We confirmed that DNA replication took place in both cases (*Figure 2E and G*). A prior study observed cohesion defects at *CEN5* in *wpl1Δ* prophase cells (*Challa et al., 2016*). We also found that deletion of *WPL1* alone caused a modest separation of sister GFP foci, but in contrast to the prior report, we observed separation of chromosome arms (*LYS2*), but not centromeres (*CEN5*) (*Figure 2F and H*). The reason for this discrepancy is unclear, but could be related to the relative distance of the *CEN5-GFP* labels from the centromere in the two studies. Importantly, *wpl1Δ* has only a minor effect on meiosis and spore viability (*Figure 2—figure supplement 2*; *Challa et al., 2016*). Furthermore, chromosome segregation fidelity in *wpl1Δ* cells is comparable to wild type (see below), indicating that cohesion is largely functional. We conclude that a critical role of Eco1 in cohesion establishment at centromeres is to counteract Wpl1, while on chromosome arms Eco1 plays an additional, essential function.

## Eco1 counteracts Wpl1-dependent cohesin destabilization during meiotic prophase

While Wpl1 promotes cohesin removal from chromosomes between prophase exit and metaphase I (*Challa et al., 2019*), paradoxically, Wpl1 levels and the slower migrating, presumed active, form

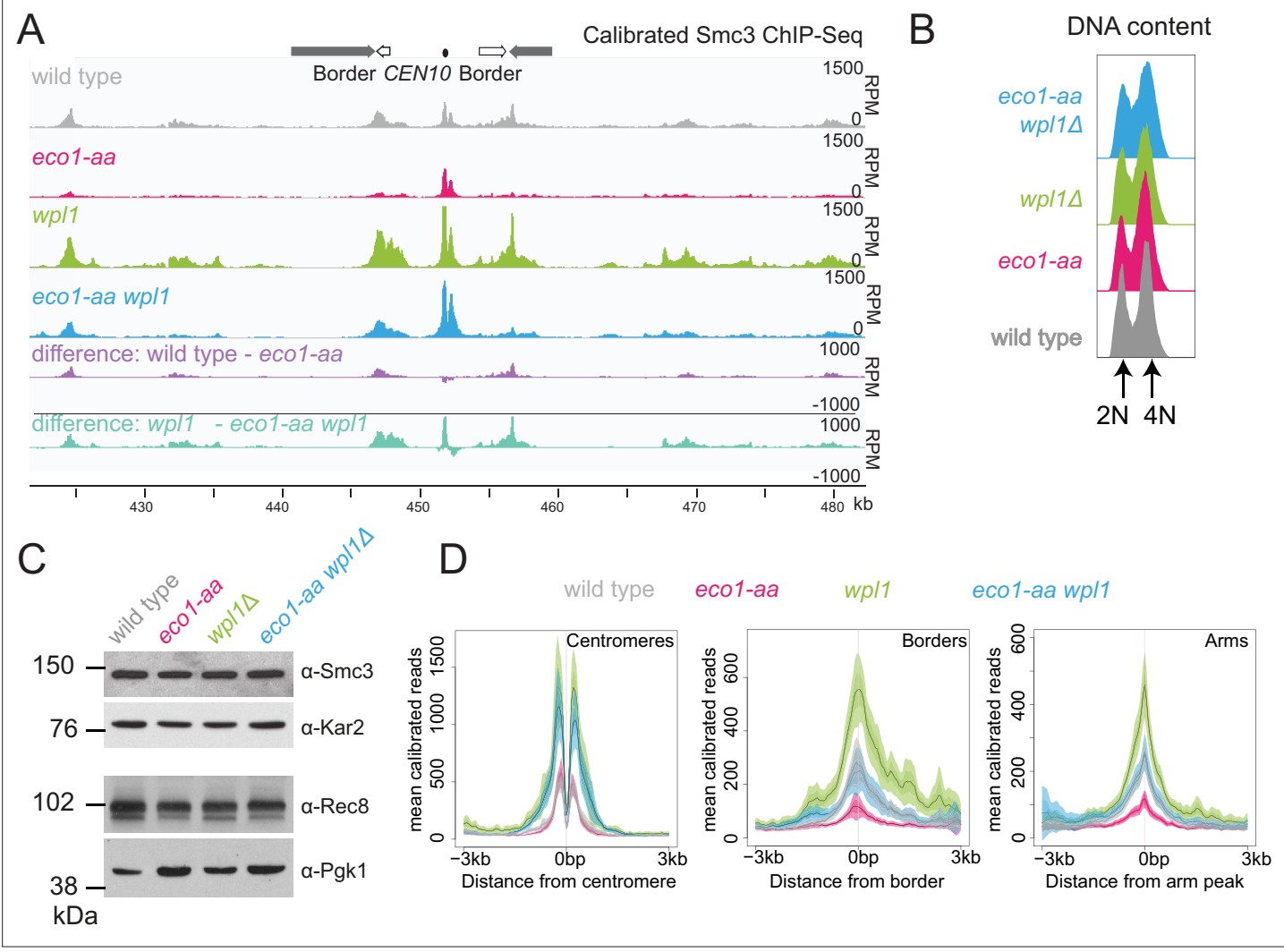

**Figure 3.** Eco1 restricts Wpl1-dependent removal of chromosomal cohesin during meiotic prophase and retains cohesin at pericentromere borders. Wpl1 globally reduces chromosomal cohesin levels, while Eco1 is required for normal cohesin levels on chromosome arms. Wild-type (AM28719), *eco1-aa* (AM28720), *wpl1Δ* (AM29750), and *wpl1Δ eco1-aa* (AM29781) anchor-away strains carrying *ndt80Δ* were harvested 6 hr after induction of sporulation. (**A**) Calibrated Smc3 ChIP-seq for a representative region surrounding *CEN10*. (**B**) Flow cytometry profiles show similar DNA content at harvesting in all cultures. (**C**) Western immunoblot with loading controls (α-Kar2 and α-Pgk1) shows comparable Smc3 (α-Smc3) and Rec8 (α-Rec8) levels in all cultures at the time of harvesting. (**D**) Mean calibrated ChIP-seq reads (line), standard error (dark shading), and 95% confidence interval (light shading) at all 16 centromeres, 32 borders, and 32 representative arm peaks.

The online version of this article includes the following source data and figure supplement(s) for figure 3:

**Source data 1.** Source images for the blot in *Figure 3C*.

**Source data 2.** Source images for the blot in *Figure 3C*.

**Source data 3.** Source images for the blot in *Figure 3C*.

**Figure supplement 1.** Eco1 retains Rec8 at pericentromere borders and antagonizes Wpl1-dependent Rec8 removal during meiotic prophase.

**Figure supplement 1—source data 1.** Source images for the blot in *Figure 3—figure supplement 1B*.

**Figure supplement 1—source data 2.** Source images for the blot in *Figure 3—figure supplement 1B*.

decline at prophase exit and are instead highest during meiotic S and prophase (*Figure 2A*). Therefore, Wpl1 may need to be counteracted by Eco1 prior to or during meiotic prophase. Calibrated ChIP-seq revealed a global increase in the levels of the cohesin subunit Smc3 on chromosomes of *wpl1Δ* prophase I cells (*Figure 3A*). We confirmed that the majority and similar numbers of cells had undergone DNA replication and that Smc3 total protein levels were comparable in all conditions

(*Figure 3B and C*). In contrast, chromosomal Smc3 was greatly reduced in *eco1-aa* genome-wide, with the exception of core centromeres, where cohesin levels were less affected (*Figure 3D*). Interestingly, inactivation of Wpl1 increased the chromosomal levels of Smc3 in *eco1-aa*, and this was particularly apparent at centromeres, where Smc3 levels in *eco1-aa wpl1Δ* were comparable to the *wpl1Δ* mutant alone (*Figure 3D*). Elsewhere, including at pericentromere borders and known chromosomal arm cohesin sites (*Paldi et al., 2020*), Smc3 levels in *eco1-aa wpl1Δ* were similar to wild type (*Figure 3D*). ChIP-seq of the meiosis-specific kleisin Rec8 in prophase I revealed a similar pattern to that of Smc3 (*Figure 3—figure supplement 1*). Although the different strains cannot be directly compared due to lack of a suitable antigen for calibration (see Materials and methods), inspection of Rec8 levels in individual strains confirmed that Eco1 is more important for Rec8 association with borders and arm sites than centromeres (*Figure 3—figure supplement 1*). Taken together, our Rec8 and Smc3 ChIP-seq show that the function of Eco1 at pericentromere borders and chromosome arms in meiotic prophase is twofold. First, Eco1 protects cohesin from Wpl1-dependent removal since cohesin levels are higher in *eco1-aa wpl1Δ* compared to *eco1-aa* cells. Second, Eco1 has an additional, Wpl1-independent function in cohesin retention at border and arm sites because cohesin levels are higher in *wpl1Δ* compared to *eco1-aa wpl1Δ* cells.

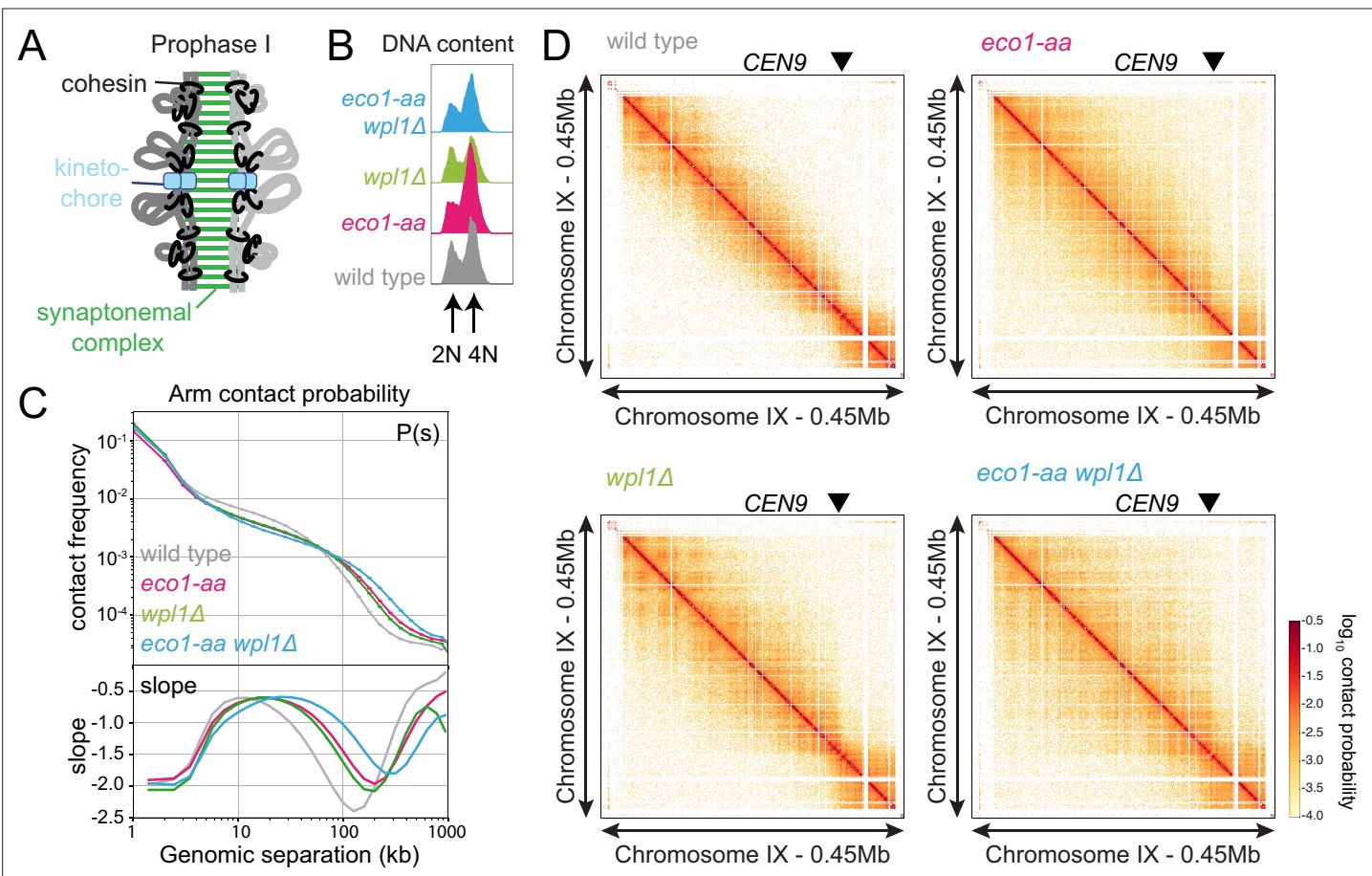

**Figure 4.** Eco1 restricts long-range chromatin interactions. Hi-C analysis of chromosome conformation in meiotic prophase of wild-type (AM28719), *eco1-aa* (AM28720), *wpl1Δ* (AM29750), and *eco1-aa wpl1Δ* (AM29781) cells carrying *ndt80Δ*. Strains were harvested 6 hr after induction of sporulation. (**A**) Schematic representation of homologous chromosomes emanating from a proteinaceous axis and depicting intra- and inter-sister chromatid cohesin in meiotic prophase. (**B**) Flow cytometry confirms similar staging in all strains. (**C**) Contact probability versus genomic distance (*P(s)*) for the indicated strains, excluding contacts across centromeres (1 kb bin; log₁₀ scale). The derivative of the *P(s)* curve (slope) plotted against genomic distance is shown below. (**D**) Contact maps (1 kb bin) show individual chromosome IX for the indicated genotypes. The arrowhead indicates the position of the centromere (*CEN9*).

# Eco1 establishes loop boundaries in meiotic prophase chromosomes and restricts long-range chromatin interactions independently from Wpl1

Meiotic prophase chromosomes are highly structured, with ordered chromatin loops emanating from linear protein axes that are zipped together by the synaptonemal complex (*Figure 4A*). To understand how Eco1 and Wpl1 define chromosome structure in meiotic prophase, we performed Hi-C 6 hr after inducing *ndt80Δ* cells to sporulate and confirmed consistent DNA content for all conditions (*Figure 4B*). Analysis of contact probability on chromosome arms as a function of genomic distance revealed an increase in long-range interactions in both *eco1-aa* and *wpl1Δ* mutants, which was further exacerbated in *eco1-aa wpl1Δ* cells (*Figure 4C*). Plotting the slope resulted in a right-ward shift of the curve maxima, which estimates the average size of the loops (*Dauban et al., 2020*; *Figure 4C*). This indicated an increase in loop size from ~10 kb in wild type to ~20 kb in *eco1-aa* or *wpl1Δ* cells and up to ~40 kb in *eco1-aa wpl1Δ* (*Figure 4C*). Heatmaps of individual chromosomes corroborated the additive increase in long-range interactions in *wpl1Δ eco1-aa* double mutants (*Figure 4D*). Further inspection revealed that spots and stripes on the Hi-C contact maps, indicative of positioned loops anchored on two or one sides, respectively, were stronger and increased in number in *wpl1Δ*, but were more diffuse in *eco1-aa*, even after *WPL1* deletion (*Figure 4D*). This suggests that loop boundaries/anchors are strengthened by *wpl1Δ* but lost in *eco1-aa*. Mirrored pileups of all 16 wild-type pericentromeres centered on the centromeres confirmed the organization of the flanking chromatin into two separate domains, indicating that centromeres act as insulators, as in mitosis (*Figure 5A*; *Paldi et al., 2020*; *Schalbetter et al., 2017*). In contrast, the absence of active Eco1 reduced insulation across centromeres (*Figure 5A*; see also top-right and bottom-left quadrants on the ratio difference maps in *Figure 5B*). The intensity of the Hi-C contact stripe protruding from centromeres increased at progressively longer distances in *eco1-aa* and *wpl1Δ eco1-aa* cells compared to wild type (*Figure 5A and B*, arrows), suggesting that Eco1 limits the extent of loop extrusion by cohesin complexes anchored at centromeres. Pileups centered on all 32 pericentromere borders (*Paldi et al., 2020*) revealed strong boundaries in wild type, which were sharper and more defined in *wpl1Δ* cells (*Figure 5C and D*). In contrast, boundaries at pericentromere borders were barely detectable in *eco1-aa* and were only partially rescued by deletion of *WPL1* (*Figure 5C and D*), suggesting that Eco1 is critical to halt loop extrusion at borders, even in the absence of Wpl1. Note that border pileups also display a second centromere-proximal stripe (arrowhead in *Figure 5C*), corresponding to loop extrusion from the centromeres that is increased in intensity in *wpl1Δ*, consistent with the centromere pileups. Both *eco1-aa* and *eco1-aa wpl1Δ* cells also exhibited reduced insulation at borders, indicating that Eco1 is also important to prevent loop extrusion across borders (*Figure 5C and D*). Consistent with the pileup analysis, examination of individual wild-type pericentromeres revealed the presence of Hi-C spots, indicative of positioned loops, between the centromere and each of the two borders, as marked by the characteristic tripartite Smc3 ChIP-seq signal (*Figure 5E*). While stronger Hi-C spots were localized with tripartite Smc3 in *wpl1Δ* cells, both features were absent in *eco1-aa* and weaker in *eco1-aa wpl1Δ* cells (*Figure 5E*). These data indicate that Wpl1 and Eco1 limit loop expansion through distinct mechanisms in meiotic prophase. While Wpl1 destabilizes loop-extruding cohesin to reduce its lifetime, Eco1 anchors cohesin at boundary sites to stabilize and position loops. We further note that Eco1-dependent loop stabilization critically defines the boundaries that demarcate the pericentromeric domain. This is consistent with the greater requirement of Eco1 to maintain cohesin association with borders and chromosomal arm sites (*Figure 3*).

## Boundaries are formed de novo in meiosis in a manner independent of DNA replication

Our findings show that Eco1 is a key determinant of loop anchors at pericentromere borders and other chromosomal boundaries in meiotic prophase cells. Since Eco1-dependent Smc3 acetylation occurs in unreplicated cells, loop anchors may form independently of DNA replication and the presence of a sister chromatid (*Figure 1E*, *Figure 1—figure supplement 2B*). Nevertheless, several observations imply that loop boundaries are established throughout meiotic S phase and prophase. First, in wild-type cells, loop extrusion progressively shapes meiotic chromosomes from G1 to prophase (*Muller et al., 2018*; *Schalbetter et al., 2019*). Second, meiotic loops are strictly dependent on Rec8 and their boundaries correspond to Rec8 peaks (*Muller et al., 2018*; *Schalbetter et al., 2019*),

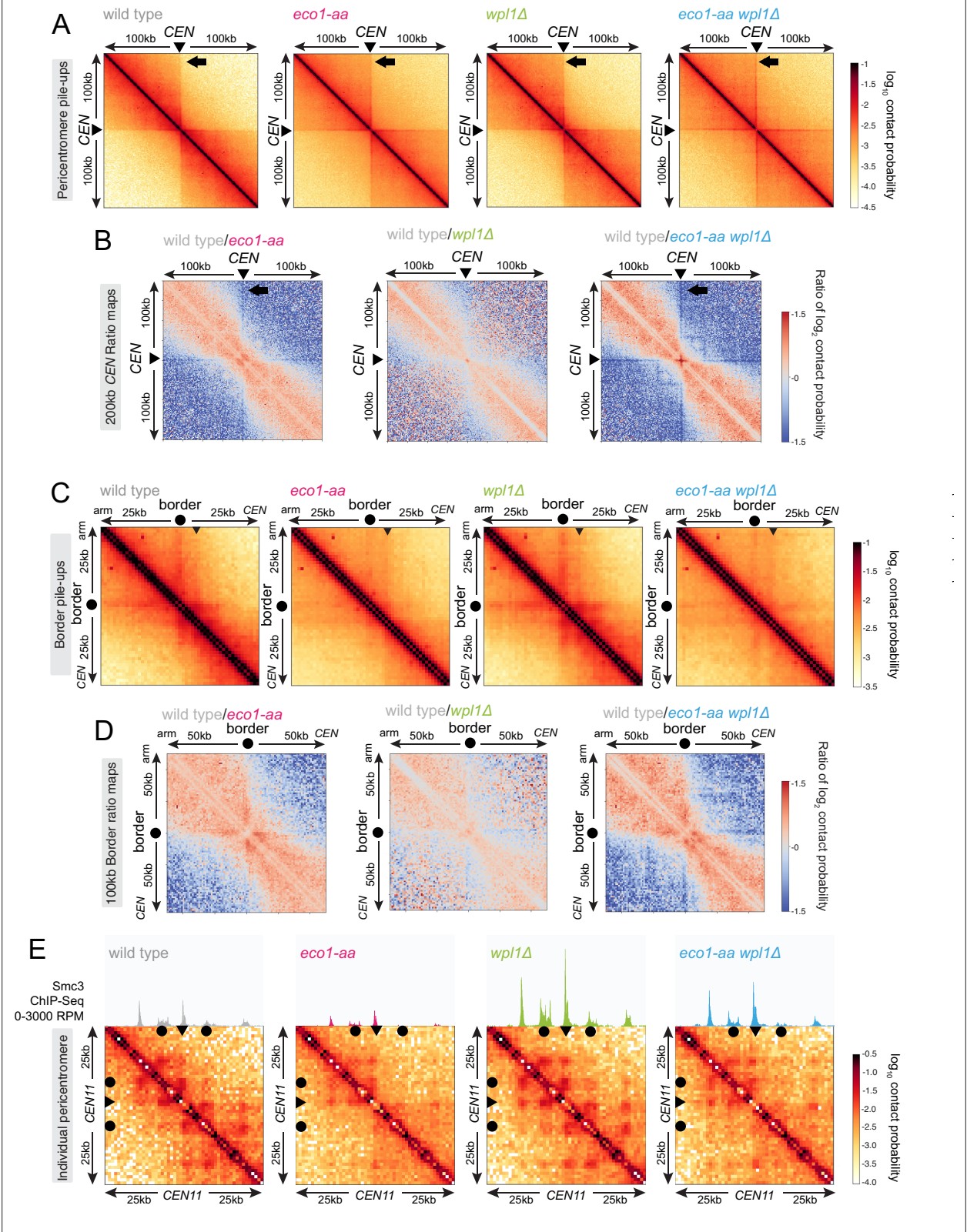

**Figure 5.** Eco1 establishes chromatin boundaries at pericentromere borders. Hi-C analysis of pericentromeric chromatin conformation in meiotic prophase of wild-type (AM28719), *eco1-aa* (AM28720), *wpl1Δ* (AM29750), and *eco1-aa wpl1Δ* (AM29781) cells. (**A**) Pileups (1 kb bin) of *cis* contacts in the 200 kb surrounding all 16 centromeres (mirrored). The arrow marks aggregate contacts between the centromeres and the region ~100 kb away on the arm of the same chromosome. (**B**) Log₂ ratio plots of centromere pileups for pairwise comparisons of *eco1-aa*, *wpl1Δ*, and *eco1-aa wpl1Δ* with the

*Figure 5 continued on next page*

*Figure 5 continued*

wild type for the 200 kb region flanking the centromeres. (**C**) Pileups (1 kb bin) of *cis* contacts in the 50 kb flanking all 32 borders. Pileups are oriented so that chromosomal arm and centromere flanks are at the upper left and lower right, respectively. Arrowhead indicates position of collated centromere-originating stripe. (**D**) Log$_2$ ratio plots of border pileups for pairwise comparisons of *eco1-aa*, *wpl1Δ*, and *eco1-aa wpl1Δ* with the wild type for the 100 kb region flanking the borders. (**E**) Contact maps (1 kb bin) of the pericentromeric region of chromosome XI (50 kb flanking *CEN11*). Calibrated Smc3 ChIP-seq signal for the appropriate genotype (from *Figure 3*) is shown above. Arrowheads mark the position of centromeres, filled circles mark the borders.

but Rec8 expression does not occur in the early G1 phase prior to meiotic entry (see *Figure 1C*). To understand if loops and loop boundaries are established during meiotic S phase and prophase, independently of DNA replication, we used Hi-C to compare chromatin conformation in wild-type and *clb5Δ clb6Δ* cells arrested before and after meiotic S phase. *clb5Δ clb6Δ* cells lack S phase cyclin-dependent kinases (CDKs) and therefore fail to initiate meiotic DNA replication (*Stuart and Wittenberg, 1998*). In either wild-type or *clb5Δ clb6Δ* cells arrested in meiotic G1 (equivalent to timepoint 0 in *Figure 1A–C*), long-range contacts were absent and no loops or loop boundaries were detected (*Figure 6A–D*, *Figure 6—figure supplement 1A and B*; *Muller et al., 2018*; *Schalbetter et al., 2019*). By prophase I, long-range contacts, loops, and loop boundaries were readily apparent in wild-type cells (*Figure 6E–H*, *Figure 6—figure supplement 1C and D*; *Muller et al., 2018*; *Schalbetter et al., 2019*). Ratio difference maps show a striking change in chromatin conformation surrounding borders and centromeres between meiotic G1 and prophase with the appearance of strong loop anchors and centromere insulation only in prophase I (*Figure 6I*, *Figure 6—figure supplement 1E*). Long-range contacts, loops, and loop boundaries arise between early G1 and prophase also in *clb5Δ clb6Δ* cells, though at lower levels compared to wild type (*Figure 6E–I*, *Figure 6—figure supplement 1C–F*). In *clb5Δ clb6Δ* cells, contact frequency in the 5–100 kb range, indicative of chromosome condensation, was only slightly increased in prophase compared to G1. However, the average loop size was slightly lower than in the wild type (~8 kb in *clb5Δ clb6Δ* vs. 10 kb in wild type; *Figure 6F*). Examination of individual pericentromeres and border pileups showed that Hi-C stripes were anchored at the pericentromere borders in *clb5Δ clb6Δ* (*Figure 6G and H*), though to a lesser extent than in wild type (see ratio maps in *Figure 6J*). The contact stripe originating from centromeres was also diminished in *clb5Δ clb6Δ* and accompanied by a loss of centromeric insulation (*Figure 6—figure supplement 1D and F*), suggesting that Clb5 and Clb6 promote centromere-anchored loop extrusion, though the mechanism is unclear. Overall, these results indicate that loop extrusion and boundary establishment occur independently of DNA replication but that DNA replication itself, or another S phase CDK-dependent process, boosts loop formation and anchoring.

## Replication-independent Smc3 acetylation defines meiotic chromosome loops

The fact that DNA replication is required neither for the formation of boundaries nor for Eco1-dependent acetylation of cohesin suggested that Eco1 could form loop anchors by directly acetylating loop-extruding cohesin complexes rather than complexes engaged in cohesion. To test this idea, we compared the effects of Eco1 on cohesin distribution and chromosome conformation in prophase-arrested unreplicated and replicated cells. Unfortunately, and for reasons that are currently unclear, we found that *clb5Δ clb6Δ eco1-aa* cells are inviable (data not shown), while *cdc6-md eco1-aa* cells were viable with no apparent growth defect. We therefore used *cdc6-md eco1-aa* cells to test the role of Eco1 in boundary formation in unreplicated prophase I cells. As expected, both *cdc6-md* and *cdc6-md eco1-aa* cells failed to undergo bulk meiotic DNA replication in contrast to wild-type and *eco1-aa* cells (*Figure 7A*, *Figure 7—figure supplement 1B*). Smc3 ChIP-seq showed that cohesin associates with the unreplicated DNA in *cdc6-md* prophase-arrested cells, though with an altered distribution compared to wild-type (*Figure 7—figure supplement 1*). Although Smc3 accumulation at centromeres in *cdc6-md* was comparable to that of wild-type cells, Smc3 peaks on chromosome arms and borders were lower and the signal was distributed between peaks in *cdc6-md* cells (*Figure 7—figure supplement 1A*). The less sharp cohesin peaks in *cdc6-md* cells suggest that DNA replication, or the presence of cohesin engaged in cohesion, limits the spreading of loop-extruding cohesin in meiosis. Inactivating Eco1 in *cdc6-md* cells further decreased both arm and pericentromeric cohesin peaks, similar to *eco1-aa* alone (*Figure 7—figure supplement 1A and D*). Therefore, Eco1 anchors

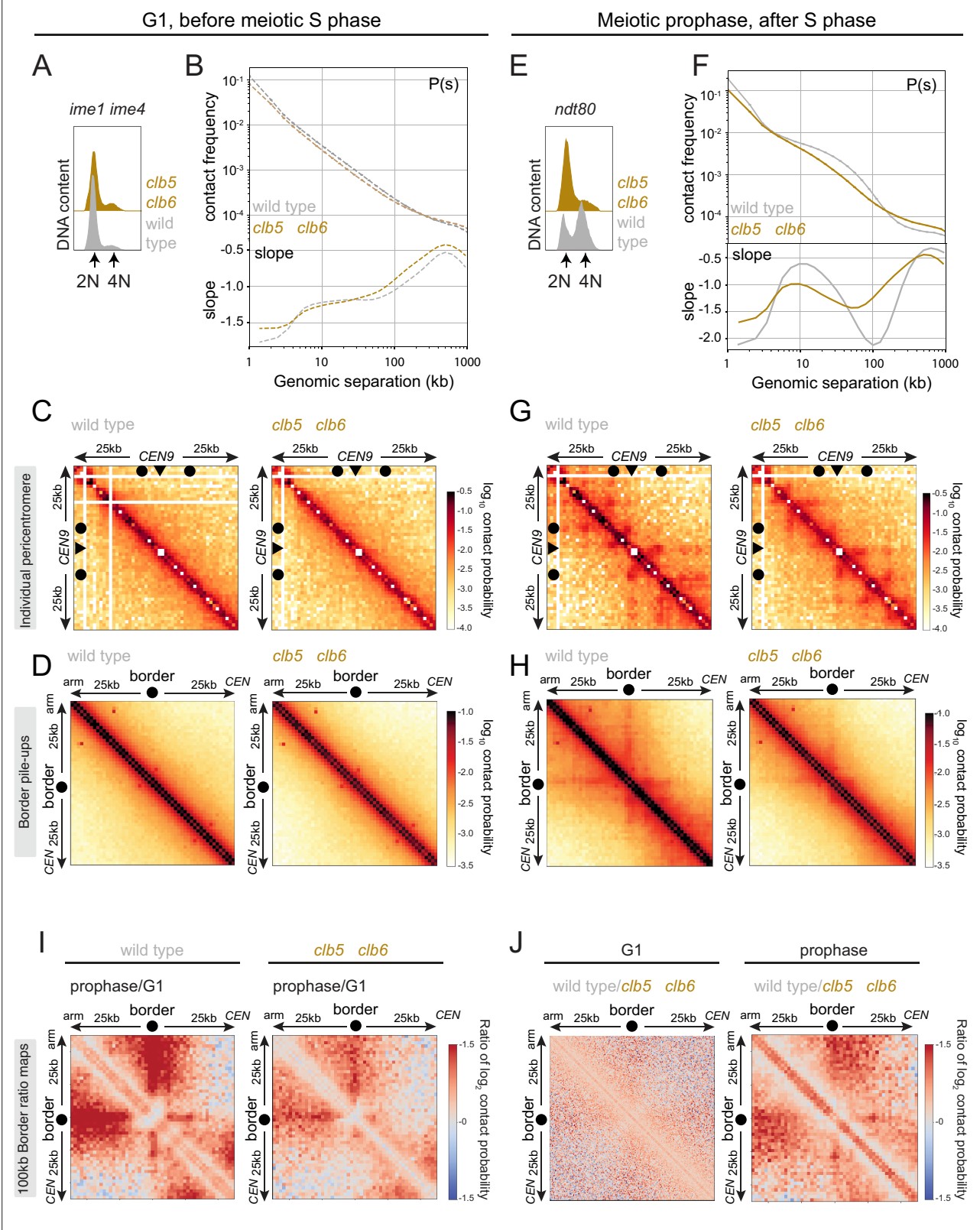

**Figure 6.** Boundaries are established at pericentromere borders independently of replication. Hi-C analysis of chromosome conformation in meiotic G1 and prophase and its dependence on DNA replication. (**A–D**) Hi-C analysis of chromosome conformation in meiotic G1 (before S phase). Wild-type (AM12145) and *clb5Δ clb6Δ* (AM26046) cells carrying *pCUP1-IME1 and pCUP1-IME4* were harvested 2 hr after induction of sporulation with no addition of CuSO₄. (**A**) Flow cytometry profiles show that DNA remains unreplicated, confirming G1 arrest. (**B**) Contact probability versus genomic distance (*P(s)*),

*Figure 6 continued on next page*

*Figure 6 continued*

excluding contacts that occur across centromeres (1 kb bin; $\log_{10}$ scale). The derivative of the *P*(*s*) curve (slope) plotted against genomic distance is shown below. (**C**) Contact maps (1 kb bin) of the pericentromeric region of the representative chromosome IX (50 kb surrounding *CEN9*). The arrowhead indicates the position of the centromere (*CEN9*), circles the position of pericentromere borders. (**D**) Pileups (1 kb bin) of *cis* contacts in the 50 kb flanking all 32 borders. Pileups are oriented so that chromosomal arm and centromere flanks are at the upper left and lower right, respectively. (**E–H**) Hi-C analysis of chromosome conformation in meiotic prophase in the absence of DNA replication. Wild-type (AM11633) and *clb5Δ clb6Δ* (AM28841) cells carrying *ndt80Δ* were analyzed as described in *Figure 4*. (**E**) Flow cytometry profiles confirm that *clb5Δ clb6Δ* fail to undergo meiotic DNA replication. (**F**) Contact probability versus genomic distance (*P*(s)), excluding contacts that occur across centromeres (1 kb bin; $\log_{10}$ scale). The derivative of the *P*(*s*) curve (slope) plotted against genomic distance is shown below. (**G**) Contact maps (1 kb bin) of the pericentromeric region of the representative chromosome IX (50 kb surrounding *CEN9*). The arrowhead indicates the position of the centromere (*CEN9*), circles represent the pericentromere borders. (**H**) Pileups (1 kb bin) of *cis* contacts in the 50 kb flanking all 32 borders. Pileups are oriented so that chromosomal arm and centromere flanks are at the upper left and lower right, respectively. (**I**) Changes in chromatin conformation at pericentromere borders between meiotic G1 and prophase. $\log_2$ ratio plots of border pileups in (**D**) and (**H**) for wild type and *clb5Δ clb6Δ* are shown. (**J**) $\log_2$ ratio plots of border pileups in (**D**) and (**H**) provide a comparison of border strength in wild type and *clb5Δ clb6Δ* in meiotic G1 (left) and prophase (right).

The online version of this article includes the following figure supplement(s) for figure 6:

**Figure supplement 1.** S phase CDKs promote loop extrusion and centromere insulation.

cohesin at least in part independently of cohesion, that is, by acting directly on loop-extruding cohesin. However, this does not rule out the possibility that Eco1 directly stabilizes both cohesin pools.

To understand how the activity of Eco1 on unreplicated chromosomes affects chromosome conformation, we next performed Hi-C in *cdc6-md* and *cdc6-md eco1-aa* cells, along with matched wild-type and *eco1-aa* controls (*Figure 7*, *Figure 7—figure supplement 2*). The contact probability *P*(*s*) curve and whole chromosome heatmaps for *cdc6-md* unreplicated prophase I cells showed an increase in long-range interactions that was more modest than that of the *eco1-aa* mutant, reflecting a small increase in the average loop size (*Figure 7B*, *Figure 7—figure supplement 2A*). This contrasts with the reduction in long-range contacts observed in *clb5Δ clb6Δ* prophase I cells (*Figure 6F*, *Figure 6—figure supplement 1C*), indicating that meiotic chromosome condensation requires a function of S phase CDKs that is independent of their role in promoting DNA replication. Combining *cdc6-md* with *eco1-aa* resulted in a further increase in long-range contacts, but not of loop size, compared to *eco1-aa* (*Figure 7B*, *Figure 7—figure supplement 2A*). Consistently, insulation across centromeres was reduced in *cdc6-md* cells to a lesser extent than in *eco1-aa* cells and the *cdc6-md eco1-aa* mutant showed an additive effect (*Figure 7—figure supplement 2B and C*). Furthermore, the characteristic stripe and dot pattern indicating the presence of loops was diminished in *cdc6-md* cells, but not as strikingly as in the *eco1-aa* and the *cdc6-md eco1-aa* mutants. This behavior can be observed on individual pericentromere maps (*Figure 7C*) but is more apparent on border pileups (*Figure 7D*) and difference maps (*Figure 7E*). The fact that formation of anchored loops in meiotic prophase I requires Eco1, but neither Cdc6 nor S phase CDKs leads to two conclusions. First, Eco1 directs the formation of cohesin-dependent boundaries and loop anchors in unreplicated cells. Second, loop anchors are further focused and strengthened in the presence of a sister chromatid. Therefore, Eco1 is likely to promote loop anchoring through cohesion-dependent and -independent mechanisms, with the latter occurring via acetylation of loop-extruding cohesin.

## Recombination prevents prophase exit in *eco1-aa* cells

We next sought to understand how boundary formation and cohesion establishment by Eco1 impact meiotic chromosome segregation. Though *eco1-aa* cells complete bulk DNA replication in meiotic S phase with similar timing to wild type (*Figure 2E and G*), only a small fraction of cells undergo nuclear divisions (*Figure 1J*). Cohesin is required for meiotic recombination, and *rec8Δ* cells undergo a recombination-dependent checkpoint arrest in meiotic prophase due to the persistence of unrepaired double-strand breaks (*Klein et al., 1999*). We found that *eco1-aa* cells similarly arrest in prophase as judged by a failure to separate spindle pole bodies (SPBs, marked by Spc42-tdTomato) and prophase exit was only modestly advanced by deletion of *WPL1* (compare *eco1-aa* to *wpl1Δ eco1-aa* cells; *Figure 8A*). This indicates that although Eco1 facilitates prophase exit by counteracting Wpl1, other Eco1 functions are also important. To determine whether activation of the recombination checkpoint prevents timely prophase exit in *eco1-aa* and *eco1-aa wpl1Δ* cells, we abolished meiotic double-strand break formation (by deletion of *SPO11*). *Figure 8A* shows that *SPO11* deletion abolished the

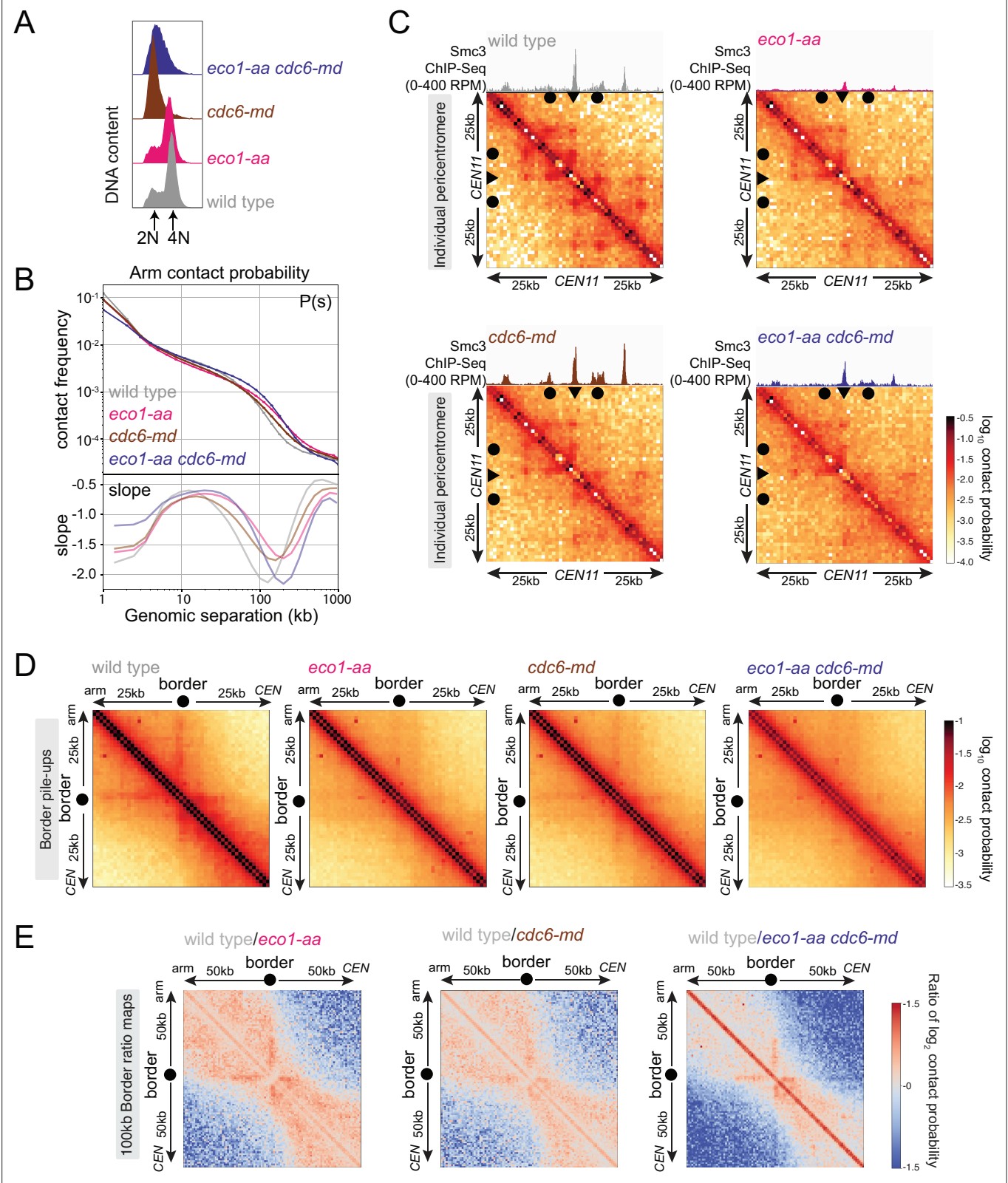

**Figure 7.** Eco1 establishes boundaries on unreplicated chromosomes. Hi-C analysis of chromosome conformation in meiotic prophase in the absence of DNA replication and/or Eco1. Wild-type (AM28719), *eco1-aa* (AM28720), *cdc6-md* (AM31003), and *eco1-aa cdc6-md* (AM30991) cells carrying *ndt80Δ* were analyzed as described in *Figure 4*. (**A**) Flow cytometry profiles confirm that *cdc6-md* and *eco1-aa cdc6-md* cells fail to undergo meiotic DNA replication while wild type and *eco1-aa* arrest with 4N DNA content. (**B**) Contact probability versus genomic distance (*P*(s)), excluding contacts that occur

*Figure 7 continued on next page*

*Figure 7 continued*

across centromeres (1 kb bin; log$_{10}$ scale). The derivative of the *P(s)* curve (slope) plotted against genomic distance is shown below. (**C**) Contact maps (1 kb bin) of the pericentromeric region of chromosome XI (50 kb flanking *CEN11*). Arrowheads mark the position of centromeres, filled circles mark the borders. (**D**) Pileups (1 kb bin) of *cis* contacts in the 50 kb surrounding all 32 borders. Pileups are oriented so that chromosomal arm and centromere flanks are at the upper left and lower right, respectively. (**E**) Log$_2$ ratio plots of border pileups for pairwise comparisons of *eco1-aa*, *cdc6-md*, and *eco1-aa cdc6-md* with the wild type for the 100 kb region flanking the borders.

The online version of this article includes the following source data and figure supplement(s) for figure 7:

**Figure supplement 1.** Cohesin distribution is altered in unreplicated cells.

**Figure supplement 1—source data 1.** Source images for the blot in *Figure 7—figure supplement 1C*.

**Figure supplement 1—source data 2.** Source images for the blot in *Figure 7—figure supplement 1C*.

**Figure supplement 1—source data 3.** Source images for the blot in *Figure 7—figure supplement 1C*.

**Figure supplement 2.** Eco1 and replication have an additive effect on restricting chromatin long-distance interactions.

prophase exit delay of both *eco1-aa* and *eco1-aa wpl1Δ* cells, confirming that Eco1 is required for satisfaction of the recombination checkpoint to allow prophase exit.

## Co-segregation of sister chromatids during meiosis I requires Wpl1 antagonism by Eco1

To understand how Eco1/Wpl1-dependent chromosome organization impacts meiotic chromosome segregation, we exploited the ability of *spo11Δ* to bypass the prophase block of *eco1-aa* cells. Note that Eco1-dependent cohesin acetylation does not require recombination (*Figure 1—figure supplement 2*). Imaging of live cells carrying a heterozygous centromere label (*CEN5*-GFP) and Spc42-tdTomato confirmed that deletion of *SPO11* permitted meiotic divisions in *eco1-aa* and *eco1-aa wpl1Δ* (*Figure 8B and C*). Our analysis of fixed prophase I cells revealed centromeric cohesion defects in *eco1-aa* cells that were rescued by *wpl1Δ* (*Figure 2F*). Similarly, live imaging of prophase I cells (one Spc42-tdTomato dot) revealed that *CEN5-GFP* foci were separated in a large fraction of *spo11Δ eco1-aa*, but not *spo11Δ eco1-aa wpl1Δ* cells (*Figure 8D*) prior to entry into meiosis I. We therefore asked whether the absence of centromeric cohesion in prophase I leads to defective meiosis I segregation. In wild-type cells, sister chromatids co-segregate in meiosis I ('reductional segregation') so that *CEN5*-GFP foci are inherited by just one of the two nuclei. This was also the case in *spo11Δ*, *wpl1Δ* and *spo11Δ wpl1Δ* cells (cells with two separated SPBs; *Figure 8C and E*, *Figure 8—figure supplement 1*). However, in ~35% of *spo11Δ eco1-aa* cells, *CEN5*-GFP foci segregated to opposite poles in meiosis I ('equational segregation'), indicating that centromeric cohesion is required for the establishment of sister kinetochore mono-orientation (*Figure 8C and E*). Consistent with its ability to restore centromeric cohesion to *eco1-aa* cells (*Figures 2F and 8D*), deletion of *WPL1* largely rescued the equational segregation of *spo11Δ eco1-aa* cells in meiosis I (*Figure 8C and E*). This supports the idea that the establishment of sister kinetochore mono-orientation requires the formation of centromeric cohesion by Eco1. Therefore, Eco1 antagonism of Wpl1 allows the establishment of centromeric cohesion to enable sister kinetochore mono-orientation.

## Sister chromatid segregation during meiosis II requires an Eco1 function distinct from Wpl1 antagonism

During wild-type meiosis II, segregation of sister chromatids to opposite poles is ensured by the pool of cohesin that persists on pericentromeres after bulk cohesin cleavage in anaphase I and which likely resides at borders. In wild-type cells, this can be visualized by sister *CEN5*-GFP foci segregation in anaphase II, resulting in their association with two of the four Spc42-tdTomato foci (*Figure 8—figure supplement 1*). Similarly, *spo11Δ, wpl1Δ,* and *spo11Δ wpl1Δ* cells segregated sister *CEN5-GFP* foci to opposite poles in meiosis II (*Figure 8C and F*, *Figure 8—figure supplement 1*). In *spo11Δ eco1-aa* cells, among the cells that enter meiosis II ~35% had already segregated sister *CEN5-GFP* foci during meiosis I (green bar in *Figure 8F*), while in a further ~40% of cells, *CEN5*-GFP were found next to a single Spc42-tdTomato focus (blue bar in *Figure 8F*), indicating defective meiosis II segregation. Moreover, defective meiosis II segregation in *eco1-aa* cells is not rescued by deletion of *WPL1*, whether or not Spo11 is present (*Figure 8F*). Therefore, Eco1 is essential for chromosome segregation during meiosis II, even in the absence of Wpl1. One potential explanation for these findings is

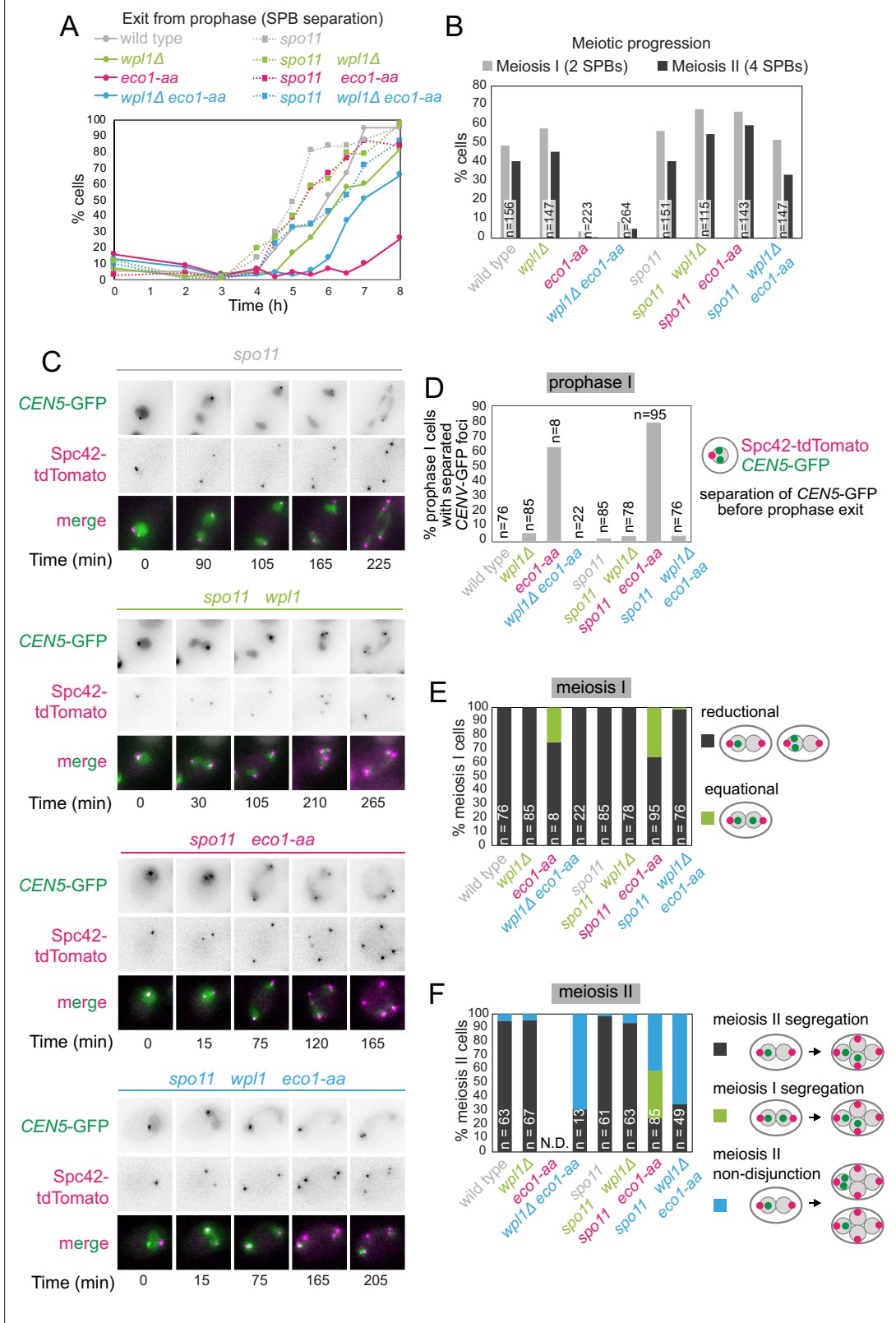

**Figure 8.** Critical Wpl1-dependent and independent roles of Eco1 in chromosome segregation during both meiosis I and II. (**A**) Meiotic double-strand break formation arrests *eco1-aa* cells in meiotic prophase. Wild-type (AM24167), *wpl1Δ* (AM24168), *eco1-aa* (AM24184), *eco1-aa wpl1Δ* (AM24169), *spo11Δ* (AM27670) *spo11Δ wpl1Δ* (AM27673), *spo11Δ eco1-aa* (AM27672), and *spo11Δ eco1-aa wpl1Δ* (AM27671) anchor-away strains carrying heterozygous *SPC42-tdTOMATO* and *CENV*-GFP were induced to sporulate. The percentage of cells with more than one Spc42-tdTomato focus was

*Figure 8 continued on next page*

*Figure 8 continued*

determined from cell populations fixed at the indicated timepoints after resuspension in SPO medium containing rapamycin. At least 100 cells were scored per timepoint. (**B–F**) Live-cell imaging of strains as in (**A**) sporulated in the presence of rapamycin. Every cell that had one Spc42-tdTomato focus at the start of the movie was scored for the duration of the movie. A small number of mitotic/dead cells (<1% for each strain) were excluded from the analysis. (**B**) Deletion of *SPO11* restores meiotic progression to *eco1-aa* cells. The percentage of cells that displayed two (meiosis I) or four (meiosis II) Spc42-tdTomato foci for each strain is shown. (**C**) Representative images of *spo11Δ* background cells undergoing the meiotic divisions. (**D**) Percentage of cells with one Spc42-tdTomato focus (prophase I) where two GFP foci were visible. Only cells that progressed to anaphase I were scored in this analysis. (**E**) Eco1 counteracts Wpl1 to allow the establishment of sister kinetochore mono-orientation. Segregation of *CEN5*-GFP foci to the same (reductional; dark gray) or opposite (equational; green) poles was scored in meiosis I (as two Spc42-tdTomato foci separate; binucleate cells). (**F**) Eco1 is required for pericentromeric cohesion, even in the absence of Wpl1. Segregation of *CEN5*-GFP foci to opposite (dark gray) or the same pole(s) (blue) was scored in meiosis II (as four Spc42-tdTomato foci separate; tetranucleate cells). Cells that had already segregated their sister *CEN5*-GFPs in meiosis I (GFP foci in two nuclei at the binucleate stage) were scored as a separate category (green).

The online version of this article includes the following source data and figure supplement(s) for figure 8:

**Figure supplement 1.** Sister kinetochore mono-orientation and pericentromeric cohesion defects in *eco1-aa* cells are not a consequence of *SPO11* deletion.

**Figure supplement 2.** Sister kinetochore mono-orientation and pericentromeric cohesion defects in *eco1-aa* cells are not caused by mislocalization of cohesin protector proteins Spo13 or Sgo1.

**Figure supplement 2—source data 1.** Source images for the blot in *Figure 8—figure supplement 2C*.

**Figure supplement 2—source data 2.** Source images for the blot in *Figure 8—figure supplement 2C*.

**Figure supplement 2—source data 3.** Source images for the blot in *Figure 8—figure supplement 2C*.

**Figure supplement 2—source data 4.** Source images for the blot in *Figure 8—figure supplement 2C*.

**Figure supplement 2—source data 5.** Source images for the blot in *Figure 8—figure supplement 2E*.

**Figure supplement 2—source data 6.** Source images for the blot in *Figure 8—figure supplement 2E*.

**Figure supplement 2—source data 7.** Source images for the blot in *Figure 8—figure supplement 2E*.

that localization of the cohesin protector protein, shugoshin (Sgo1), or the meiotic protein Spo13, which is also required for cohesin protection during meiosis I (*Galander et al., 2019b*; *Galander et al., 2019a*; *Katis et al., 2004*; *Lee et al., 2004*), may require Smc3 acetylation. However, ChIP-qPCR revealed that both Sgo1 and Spo13 localize to chromosomes in *eco1-aa* cells (*Figure 8—figure supplement 2*). Both proteins follow a similar pattern to Rec8, showing reduced chromosomal association in *eco1-aa* cells that is rescued by *WPL1* deletion (*Figure 8—figure supplement 2*), consistent with a requirement for cohesin for the chromosomal association of Sgo1 and Spo13 (*Galander et al., 2019b*; *Kiburz et al., 2005*). Therefore, a failure in cohesin protection is unlikely to be the cause of meiosis II missegregation in *eco1-aa* cells. Since *wpl1Δ* rescues the loss of cohesin on pericentromeric borders in *eco1-aa* cells, but not the anchoring of loops, it is likely that Eco1 enables accurate meiosis II segregation by anchoring cohesin at chromatin boundaries to generate robust cohesion at pericentromere borders.

## Mutation of Smc3-K112,113 results in meiotic lethality

In mitotically growing cells, Eco1 protects cohesin from Wpl1 by acetylation of Smc3 residues K112 and K113 (*Rolef Ben-Shahar et al., 2008*; *Rowland et al., 2009*; *Unal et al., 2008*). To determine whether Smc3-Ac similarly allows cohesion establishment by protecting from Wpl1 and/or confers chromatin boundary function in meiosis, we analyzed a non-acetylatable *smc3-K112,113R* mutant. To support mitotic growth, cells carried wild-type *SMC3* under the *CLB2* promoter, which is repressed in meiosis (homozygous *pCLB2-3HA-SMC3*; henceforth Smc3 *m*eiotic *d*epletion, *smc3-md*), together with heterozygous *smc3-K112,113R,* or *SMC3* as a control, at an ectopic locus expressed from the endogenous promoter (*Figure 9—figure supplement 1A*). Smc3 levels in *smc3-md* cells without ectopic expression were largely repressed, though low levels of residual Smc3 were detectable (*Figure 9—figure supplement 1B and C*). Levels of heterozygously produced Smc3 and Smc3-K112,113R were approximately half that of Smc3 in wild-type cells (*Figure 9—figure supplement 1B and C*). Ectopic *SMC3* expression rescued the inviability of *smc3-md* spores while *smc3-K112,113R* expression did not, indicating that Smc3-Ac is essential for meiosis (*Figure 9—figure supplement 1D*). To determine whether Smc3-K112,113R localizes normally to chromosomes and whether it is susceptible to destabilization by Wpl1, we performed calibrated Smc3 ChIP-seq in prophase I cells

where either Smc3 or Smc3-K112,113R were heterozygously produced and in the presence and absence of Wpl1 (*Figure 9A–D*). Flow cytometry confirmed similar meiotic progression (*Figure 9A*), and Western blotting showed that total cellular Smc3 levels were comparable (*Figure 9B*). Like *eco1-aa* (*Figure 3A–D*), *smc3-K112,113R* reduced chromosomal cohesin levels genome-wide, but had a milder effect at centromeres (*Figure 9C and D*). *WPL1* deletion increased Smc3 levels in *smc3-K112,113R* cells (*Figure 9C and D*), but to a lesser extent than in the *eco1-aa* background (*Figure 3A*). Although the reasons for this are unclear, the finding that these residues are involved in binding DNA (*Shi et al., 2020*; *Bauer et al., 2021*) suggests that the amino acid substitutions may themselves perturb cohesin loading or translocation, in addition to preventing acetylation by Eco1. Nevertheless, although introduction of the *smc3-K112,113R* mutation into *wpl1Δ* cells only slightly reduced Smc3 levels at centromeres, it greatly reduced Smc3 levels at pericentromeric borders and arm sites (*Figure 9C and D*, compare *wpl1Δ* and *smc3-K112,113R wpl1Δ*). Therefore, like Eco1, Smc3-Ac is more important for retention of cohesin at pericentromere borders and arm sites than at centromeres, suggesting a role in boundary formation.

We asked whether Smc3-Ac underlies the functional effects of Eco1 in meiosis by conferring both cohesion establishment and boundary function. First, we scored sister *CEN5*-GFP or *LYS2*-GFP foci separation as cells progressed into a prophase arrest and found that *smc3-K112,113R* cells exhibited similar cohesion defects to cells depleted of Smc3 (*Figure 9—figure supplement 2A–F*). In both *smc3-K112,113R* and *smc3-md* cells, the cohesion defect at centromeres was more modest than that of *eco1-aa* cells (*Figure 9—figure supplement 2F*), potentially reflecting incomplete meiotic depletion of Smc3 when placed under the *CLB2* promoter (*Figure 9—figure supplement 1B and C*). Next, we used live-cell imaging to assay the requirement for Smc3-Ac in prophase exit and meiosis I and II chromosome segregation. Exit from prophase was impaired in *smc3-md* and, to a lesser extent, *smc3-K112,113R* cells, but in both cases was overcome by deletion of *SPO11* (*Figure 9E*, *Figure 9—figure supplement 3A*), indicating that Smc3-Ac is required for satisfaction of the recombination checkpoint, like Eco1. Analysis of sister chromatid segregation in meiosis I in the *spo11Δ* background (*Figure 9F*, *Figure 9—figure supplement 3B*) revealed that ~10–15% of *smc3-md* and *smc3-K112,113 R* cells aberrantly segregated sister chromatids to opposite poles during meiosis I. In the case of *smc3-K112,113R*, but not *smc3-md*, this equational meiosis I segregation was rescued by deletion of *WPL1* (*Figure 9F*). Sister chromatid segregation during meiosis II was also greatly impaired in both *smc3-md* and *smc3-K112,113R* cells, but in neither case was it rescued by *WPL1* deletion (*Figure 9G*, *Figure 9—figure supplement 3C*). We note that *smc3-K112,113R* and *smc3-md* have a lesser effect on meiosis I sister chromatid segregation and centromeric cohesion as compared to *eco1-aa* (compare *Figure 2F* with *Figure 9—figure supplement 2A*). In contrast, meiosis II segregation was similarly defective in *eco1-aa*, *smc3-md* and *smc3-K112,113R* cells (*Figure 8F* and *Figure 9G*). The most likely explanation for this difference is that pre-meiotic expression of *smc3-md* leads to persistence of functional Smc3 (*Figure 9—figure supplement 1C*), which is preferentially loaded at centromeres to partially establish cohesion at this site. However, *WPL1* deletion similarly affected both meiosis I and II segregation in *eco1-aa* and *smc3-K112,113R* cells. We conclude that Eco1-dependent Smc3-Ac functionally organizes meiotic chromosomes for their segregation. At centromeres, Eco1-dependent Smc3 acetylation counteracts Wpl1 to direct sister kinetochore mono-orientation and thereby enforce sister chromatid co-segregation in meiosis I. Eco1 acetylation of Smc3 is also essential for prophase exit and sister chromatid segregation in meiosis II, even in the absence of Wpl1, likely due to its requirement for cohesin anchoring to establish chromatin boundaries.

## Discussion
### Cohesin regulators organize functionally distinct chromosomal domains
We have shown how cohesin regulators remodel chromosomes into functionally distinct domains to allow for the unique events of meiosis. We find that the control of loop formation and cohesion establishment by Eco1 and Wpl1 allows centromeres, pericentromeres, and chromosome arms to adopt region-specific functions in sister kinetochore co-orientation, cohesion maintenance, and recombination. These functions of Eco1 and Wpl1 that we uncovered in meiosis may also explain how other chromosome domains are established to support other genomic functions, including localized DNA repair and control of gene expression.

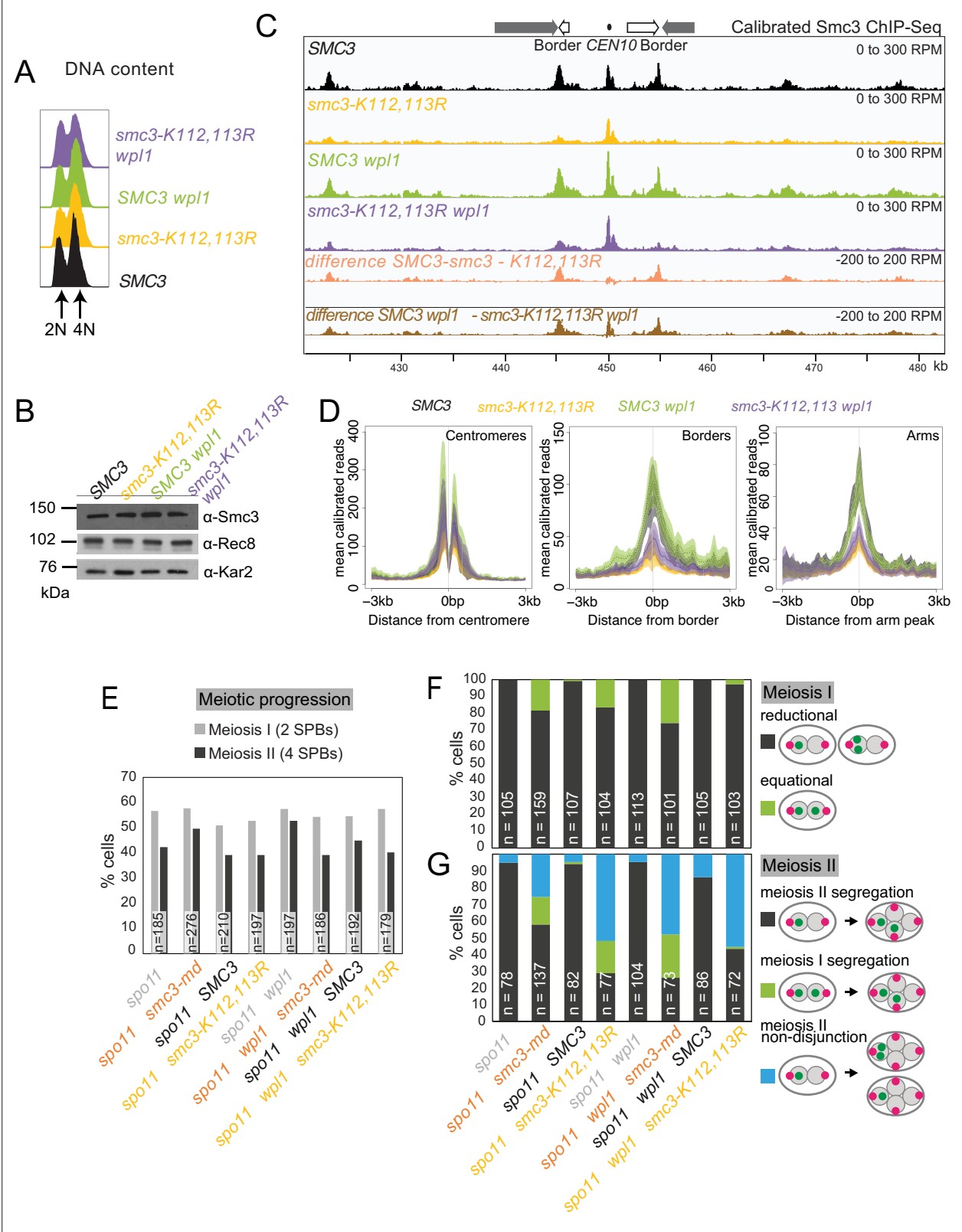

**Figure 9.** Smc3 acetylation is essential for meiosis. (**A–D**) *smc3-K112,113R* leads to a global reduction in chromosomal Smc3 levels, which is only partially restored by *WPL1* deletion. *SMC3* (AM29315), *smc3-K112,113R* (AM29316), *SMC3 wpl1Δ* (AM30310), and *smc3-K112,113R wpl1Δ* (AM30311) strains carrying *ndt80Δ* were harvested 6 hr after induction of sporulation. (**A**) Flow cytometry profiles show similar DNA content at harvesting in all cultures. (**B**) Western immunoblot with Kar2 loading control (α-Kar2) shows comparable Smc3 (α-Smc3) and Rec8 (α-Rec8) levels in all cultures at the

*Figure 9 continued on next page*

*Figure 9 continued*

time of harvesting. (**C**) Calibrated Smc3 ChIP-seq for a representative region surrounding *CEN10*. (**D**) Mean calibrated ChIP-seq reads (line), standard error (dark shading), and 95% confidence interval (light shading) at all 16 centromeres, 32 borders, and 32 flanking arm sites. (**E–G**) Smc3-Ac is required to ensure co-segregation of sister chromatids in meiosis I and accurate meiosis II chromosome segregation. Meiotic progression (**E**) and meiosis I (**F**) and II (**G**) chromosome segregation were scored after live-cell imaging as in *Figure 6* (**B–E**). Strains used were *spo11Δ* (AM30238), *spo11Δ smc3-md* (AM30240), *spo11Δ SMC3* (AM30242), *spo11 smc3-K112,113R* (AM30244), *spo11Δ wpl1Δ* (AM30234), *spo11Δ wpl1Δ smc3-md* (AM30235), *spo11Δ wpl1Δ SMC3* (AM30655), and *spo11Δ wpl1Δ smc3-K112,113R* (AM30237).

The online version of this article includes the following source data and figure supplement(s) for figure 9:

**Source data 1.** Source images for the blot in *Figure 9B*.

**Source data 2.** Source images for the blot in *Figure 9B*.

**Source data 3.** Source images for the blot in *Figure 9B*.

**Figure supplement 1.** A system to express Smc3-K112,113R in meiosis.

**Figure supplement 1—source data 1.** Source images for the blot in *Figure 9C*.

**Figure supplement 1—source data 2.** Source images for the blot in *Figure 9C*.

**Figure supplement 2.** Establishment of meiotic cohesion requires Smc3 acetylation.

**Figure supplement 2—source data 1.** Source images for the blot in *Figure 9B and E*.

**Figure supplement 2—source data 2.** Source images for the blot in *Figure 9B and E*.

**Figure supplement 3.** Prevention of meiotic recombination allows *smc3-K112,113R* cells to efficiently exit prophase.

## Loop anchoring allows the formation of specific chromosomal boundaries

Eco1 associates with replication factors and is proposed to couple cohesion establishment to DNA replication by traveling with replication forks (*Ivanov et al., 2018*; *Ladurner et al., 2016*; *Lengronne et al., 2006*; *Song et al., 2012*). In mitotically dividing yeast cells, Smc3 acetylation is largely coupled to DNA replication (*Rolef Ben-Shahar et al., 2008*); however, during meiotic S phase, substantial Smc3 acetylation, and anchoring of chromatin boundaries, occurs even in the absence of DNA replication. The anchored loops that form on unreplicated chromatin require Eco1 similarly to those on replicated chromatin. Cohesin acetylation also occurs without DNA replication in mammalian G1 cells, with a preference for STAG1-containing cohesin complexes (*Alomer et al., 2017*; *Wutz et al., 2020*). Therefore, Smc3-Ac can exist independently of cohesion establishment. Indeed, Smc3-Ac per se is not critical for cohesion since mitotic *eco1Δ wpl1Δ* yeast cells build sufficient cohesion to support viability. How Eco1 gains access to cohesin that is not associated with replication forks remains to be understood, but it is interesting to speculate that specialized cohesin subunits, for example, STAG1 in mammals or Rec8 in yeast, may allow cohesin targeting by Eco1 independent of the replication machinery.

In addition to Wpl1 antagonism, Eco1 and Smc3-Ac confer a more fundamental role in chromosomal loop positioning that we find to be indispensable for meiosis. We envisage that early in meiosis, prior to DNA replication, Scc2-Scc4-cohesin complexes load onto chromosomes and begin to extrude loops (*Figure 10*). Loaded cohesin, and loops, has a limited lifetime due to the removal of unacetylated cohesin by Wpl1. However, a fraction of cohesin engaged in *cis*-looping is acetylated by Eco1, with two consequences. First, Smc3-Ac blocks cohesin's Wpl1-dependent release. Second, Smc3-Ac anchors cohesin at the base of loops, preventing loop migration and restricting extrusion, resulting in the stable positioning of moderately sized loops. The absence of Wpl1 increases the lifetime of loop-extruding cohesin complexes, resulting in loop extension (*Figure 4*; *Costantino et al., 2020*; *Dauban et al., 2020*; *Haarhuis et al., 2017*). The combined absence of both Eco1 and Wpl1 results in mobile cohesin with extended lifetimes on chromosomes and untempered loop-extruding activity, leading to long, poorly positioned loops. Therefore, the role of Eco1 in anchoring cohesin is Wpl1 independent and is essential for meiosis. The positioning activity of Eco1 also facilitates cohesion establishment along chromosome arms by anchoring cohesin complexes that are engaged in linking the two sister chromatids ('cohesive cohesin'). Whether it does this directly by acetylating cohesive cohesin to prevent its migration or indirectly by generating *cis*-loops that form a barrier to translocation of cohesive cohesin is unclear. Our finding that the converse is also true, that is, that loop boundaries are increased in focus in the presence of a sister chromatid, together with the fact that functional

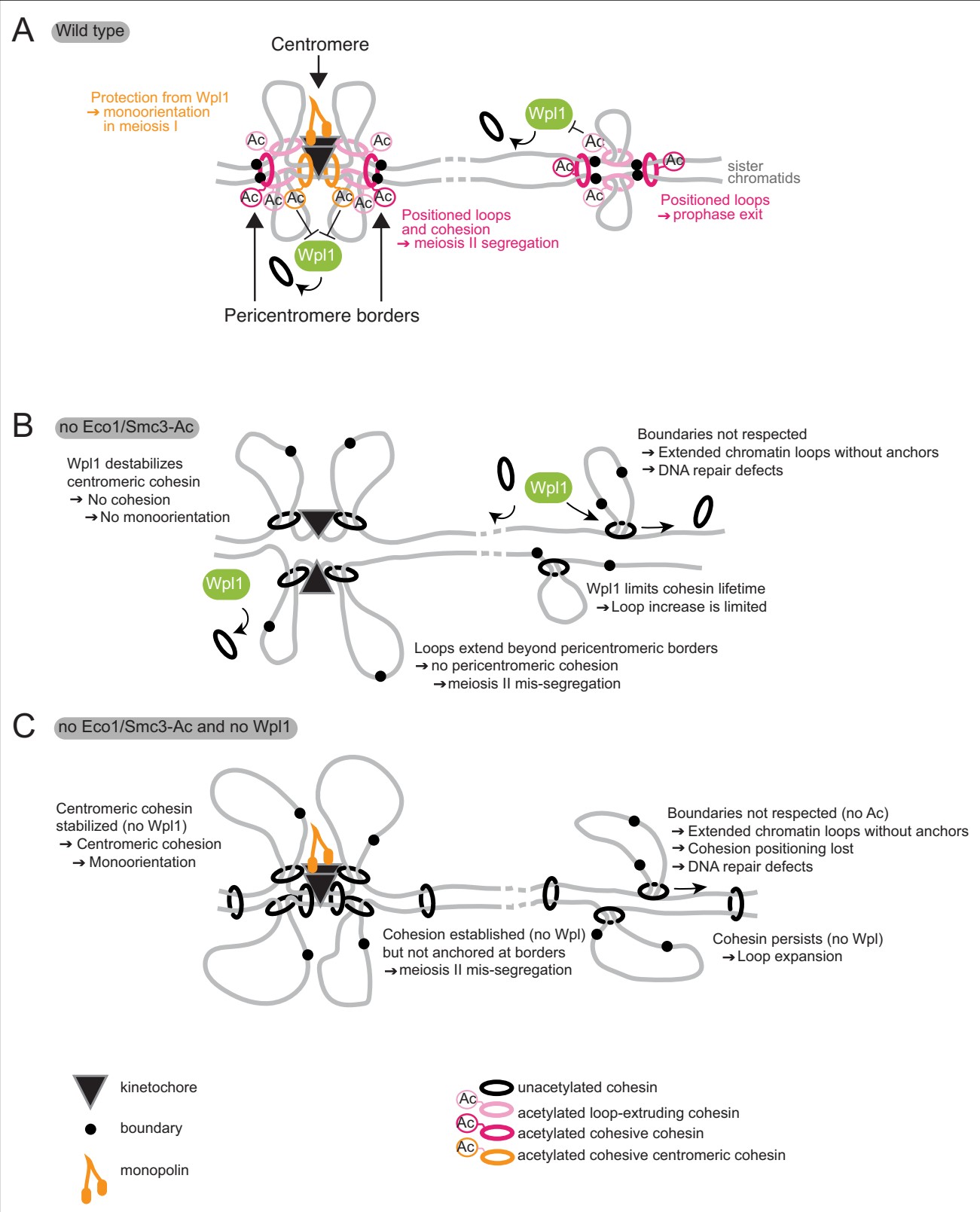

**Figure 10.** Model for Eco1 and Wpl1 roles in meiosis. (**A**) In wild-type cells (top panel), Eco1 cohesin acetylation is essential in three meiotic processes: it protects centromeric cohesin from Wpl1-mediated release, allowing sufficient cohesion to be built to establish mono-orientation; it positions DNA loops along chromosome arms to promote recombination and prophase exit; and it positions loops and cohesion at pericentromeric borders to guide correct sister chromatid segregation in meiosis II. (**B**) In the absence of Eco1 or Smc3-Ac, boundaries are not respected and more cohesin complexes

*Figure 10 continued on next page*

*Figure 10 continued*

are released from DNA due to the action of Wpl1, with detrimental effects on recombination and meiosis I and II segregation. (**C**) In the absence of both Eco1 or Smc3-Ac and Wpl1, cohesion is partially restored, specifically at the centromere, but loop boundaries are not, leading to the formation of long unpositioned loops that are not able to support prophase recombination and meiosis II segregation.

pericentromeric cohesion in meiosis II also requires Eco1, suggests that cohesive cohesin complexes are also directly positioned by acetylation, At centromeres, Smc3-Ac, which blocks the destabilizing activity of Wpl1, is sufficient for cohesion establishment, likely due to enhanced cohesin loading or specialized anchoring mechanisms at this site. A recent report indicates that Eco1-dependent Smc3 acetylation also establishes loop positioning in S phase of yeast mitotic growth (*Bastié et al., 2021*). Exactly how acetylation affects cohesin enzymology awaits detailed biochemical analysis. However, recent reports investigating the relationship between molecular structure and loop-extruding activity of human cohesin identified the residues acetylated by Eco1 as part of the surface that binds DNA in the cohesin 'clamp' structure. The clamp is one of the conformational transitions occurring in the part of the cohesin cycle required for the loop extrusion step (*Bauer et al., 2021*; *Shi et al., 2020*).

## Establishment of functional chromosomal units for meiotic recombination

Our studies on meiosis, where chromosomal domains must be defined to lose cohesin at chromosome arms in meiosis I and at pericentromeres in meiosis II and to restrict recombination to chromosome arms, provide a unique opportunity to dissect the functional importance of loop positioning. We found that Eco1 and Smc3-Ac are critical for the completion of meiotic recombination to allow exit from meiotic prophase. Restoration of cohesin levels in Eco1-deficient cells by removal of Wpl1 was insufficient to support DNA repair and prophase exit, or for chromosomal arm cohesion. We speculate that loss of positioned chromatin loops causes defects in the repair of double-strand breaks, leading to a delay in prophase exit. Eco1-dependent cohesin anchoring may promote homology search, be required to reel broken ends within loops to chromosome axes for repair, and/or ensure that loops are positioned in register with the homologous chromosome. Although positioned loops are readily detectable during meiotic prophase in budding yeast (*Schalbetter et al., 2019*; *Figure 4*), loops appear less positioned in mouse, rhesus monkey, human, and mouse spermatocytes (*Alavattam et al., 2019*; *Patel et al., 2019*; *Wang et al., 2019*; *Zuo et al., 2021*). Whether this underlies biological differences in mammals or a technical limitation due to the larger genome size remains unclear.

Eco1 is present beyond S phase into prophase, suggesting that it may play an active role in loop repositioning and/or the anchoring of new loops during homologous recombination. However, Eco1 is degraded during prophase, after which only existing loop anchors and positioned cohesion will persist.

## Centromeric cohesion directs sister chromatid co-segregation during meiosis I

Centromeres, differently from chromosome arms, require Eco1 for establishing cohesion only in the presence of Wpl1. Our live-cell imaging revealed that the loss of centromeric cohesion in *eco1-aa* cells is accompanied by the aberrant segregation of sister chromatids to opposite poles in meiosis I, which was also rescued by Wpl1 inactivation. In Smc3-depleted and *smc3-K112,113R* cells, likely due to residual *SMC3* expression from the *CLB2* promoter, the centromeric cohesion defect was more modest and segregation of sister chromatids to opposite poles was less frequent, but was nevertheless rescued by Wpl1 inactivation. Although we cannot currently rule out the possibility that Eco1 has an additional substrate that confers its function in sister kinetochore co-orientation as has been suggested for fission yeast (*Kagami et al., 2017*), the simplest explanation is that acetylation of Smc3 by Eco1 counteracts Wpl1 at centromeres allowing the establishment of centromeric cohesion, which in turn facilitates kinetochore fusion via the monopolin complex (reviewed in *Duro and Marston, 2015*). Interestingly, budding yeast lacking Rec8 cohesin do not show co-orientation defects (*Monje-Casas et al., 2007*; *Ciosk et al., 2000*), raising the possibility that mitosis-like, Scc1-containing cohesin complexes confer the co-orientation function of cohesin at centromeres. It will be of great

interest to understand how monopolin and cohesin-mediated co-orientation pathways intersect to ensure proper meiosis I chromosome segregation.

## Pericentromere boundaries define persistent cohesion at meiosis II

During meiosis, cohesion on chromosome arms requires Eco1 and Smc3-Ac even in the absence of Wpl1. This argues against the idea that the key function of Smc3-Ac in cohesion is to antagonize Wpl1. Instead, we propose that the ability of Eco1 to reinforce chromatin boundaries is critical for cohesion during meiosis. This phenomenon is most apparent at pericentromere borders, where centromere-loaded cohesin is trapped by convergent genes in mitosis (*Paldi et al., 2020*). In meiosis II, cells rely entirely on cohesin at pericentromere borders to hold sister chromatids together. We now show that Eco1 is required to retain cohesin at these sites and, consequently, to establish the boundaries that structure pericentromeres during meiosis. As a result, sister chromatids undergo random meiosis II segregation in cells lacking Eco1, whether or not Wpl1 is present. This can explain why cohesin anchoring is essential in meiosis but not in mitosis, where cohesin is present also along chromosome arms. Understanding how Eco1-dependent Smc3 acetylation and convergent genes together trap cohesin to elicit loop anchoring and cohesion at pericentromere borders is an important question for the future.

The critical importance of cohesion and loop positioning for chromosome segregation has important implications for meiosis in human eggs, where the events of prophase and the meiotic divisions are temporally separated by years. It will be interesting to understand whether Esco1/2 remain active in mammalian eggs and whether this is important to safeguard cohesin during the female reproductive lifespan.

## Importance of cohesin anchoring

A universal feature of cohesin-dependent chromosome organization is the existence of boundary elements that position both loops and cohesion. However, how these boundaries are established and the functional importance of cohesin anchoring were unclear. We found that Eco1-dependent cohesin acetylation confers boundary recognition and cohesin anchoring. The importance of loop anchoring was revealed by their requirement for proper recombination and pericentromeric cohesion in meiotic chromosome segregation. Our demonstration of the importance of cohesin anchoring for fundamental chromosomal processes provides a framework for understanding how chromatin folding into defined domains maintains and propagates the genome.

# Materials and methods

**Key resources table**

| Reagent type (species) or resource | Designation | Source or reference | Identifiers | Additional information |
|---|---|---|---|---|
| Strain, strain background (*Saccharomyces cerevisiae*) | NCBITaxon: 4932 | This paper | Yeast strains; RRID:SCR_003093 | *Supplementary file 1* |
| Strain, strain background (*Schizosaccharomyces pombe*) | NCBITaxon: 4896 | This paper | Yeast strains; RRID:SCR_010536 | *Supplementary file 1* |
| Antibody | Anti-HA 12CA5 (mouse monoclonal) | Roche | Cat# 11666606001; RRID:AB_514506 | WB (1:1000), ChIP |
| Antibody | Anti-HA 11 (mouse monoclonal) | BioLegend | Cat# MMS-101R; RRID:AB_291262 | WB (1:1000) |
| Antibody | Anti-FLAG M2 (mouse monoclonal) | Sigma | Cat# F1804; RRID:AB_262044 | WB (1:1000), ChIP |
| Antibody | Anti-Pgk1 (rabbit polyclonal) | Marston lab stock | n/a | WB (1:10,000) |
| Antibody | Anti-Kar2 (rabbit polyclonal) | Marston lab stock | n/a | WB (1:10,000) |
| Antibody | Anti Smc3-K112,113Ac (rabbit polyclonal) | Marston lab stock | n/a | WB (1:1000) |
| Antibody | Anti-Rec8 (rabbit polyclonal) | Marston lab stock | n/a | WB (1:15,000), ChIP |

*Continued on next page*

*Continued*

| Reagent type (species) or resource | Designation | Source or reference | Identifiers | Additional information |
|---|---|---|---|---|
| Antibody | Anti-Smc3 (rabbit polyclonal) | Marston lab stock | n/a | WB (1:1000), ChIP |
| Antibody | Anti-mouse HRP (sheep monoclonal) | GE Healthcare | Cat# NXA931; RRID:AB_772209 | WB (1:10,000) |
| Antibody | Anti-rabbit (donkey monoclonal) | GE Healthcare | Cat# NA934; RRID:AB_772206 | WB (1:10,000) |
| Antibody | Anti-tubulin alpha (rat monoclonal) | Bio-Rad AbD Serotec | Cat# MCA78G; RRID:AB_325005 | IF (1:50) |
| Antibody | Anti-rat FITC antibody (donkey polyclonal) | Jackson ImmunoResearch | Cat# 712-095-153; RRID:AB_325005 | IF (1:100) |
| Recombinant DNA reagent | AMp1342 (plasmid) | This paper | YIplac128-SMC3 | *LEU2* integration plasmid carrying *SMC3* |
| Recombinant DNA reagent | AMp1392 (plasmid) | This paper | YIplac128-smc3-K112R,K113R | *LEU2* integration plasmid carrying *smc3-K112R,K113R* |
| Sequence-based reagent | Primers | This paper | qPCR primers | **Appendix 1—table 1** |
| Peptide, recombinant protein | Zymolyase | AMS Biotechnology | Cat# 120491-1 | |
| Peptide, recombinant protein | Glusolase | PerkinElmer | Cat# NEE154001EA | |
| Peptide, recombinant protein | RNAse A | Amresco | Cat# 0675–250MG | |
| Peptide, recombinant protein | Proteinase K | Invitrogen | Cat# 25530015 | |
| Peptide, recombinant protein | Klenow (exo-) enzyme | NEB | Cat# M0212S | |
| Peptide, recombinant protein | Phusion High-Fidelity DNA Polymerase | NEB | Cat# F530L | |
| Peptide, recombinant protein | Klenow fragment Dpol I | NEB | Cat# M0210L | |
| Peptide, recombinant protein | T4 DNA ligase | Invitrogen | Cat# 15224090 | |
| Peptide, recombinant protein | T4 DNA Polymerase | NEB | Cat# M0203 | |
| Peptide, recombinant protein | T4 Polynucleotide Kinase | NEB | Cat# M0201 | |
| Commercial assay or kit | QuikChange XL Site-directed mutagenesis kit | Agilent | Cat# 200516 | |
| Commercial assay or kit | SYBR GreenER | Invitrogen | Cat# 11762500 | |
| Commercial assay or kit | Luna Universal qPCR Master Mix | NEB | Cat# M3003X | |
| Commercial assay or kit | Quick Ligation kit | NEB | Cat# M2200L | |
| Commercial assay or kit | 2100 Bioanalyzer High Sensitivity DNA kit | Agilent | Cat# 5067-4626 | |
| Commercial assay or kit | Qubit HS DNA assay kit | Thermo Scientific | Cat# Q32854 | |
| Commercial assay or kit | ECL SuperSignal West Pico chemiluminescence kit | Thermo Scientific | Cat# 34580 | |
| Commercial assay or kit | SuperSignal West Femto chemiluminescence kit | Thermo Scientific | Cat# 34094 | |
| Chemical compound, drug | Complete EDTA-free protease inhibitor cocktail tablets | Roche | Cat# 1873580001 | |
| Chemical compound, drug | NEB Buffer 2 | NEB | Cat# B7202 | |
| Chemical compound, drug | NEB Buffer 3.1 | NEB | Cat# B7203S | |
| Chemical compound, drug | 0.4 mM biotin-14-dCTP | Invitrogen | Cat# 19518018 | |

*Continued on next page*

Continued

| Reagent type (species) or resource | Designation | Source or reference | Identifiers | Additional information |
|---|---|---|---|---|
| Chemical compound, drug | Phenol:chloroform:isoamyl alcohol 25:24:1 | Sigma | Cat# P3803 | |
| Software, algorithm | ImageJ software version 2.0.0-rc-43/1.51g | National Institutes of Health | https://imagej.nih.gov/ij/; RRID:003070 | |
| Software, algorithm | Integrated Genome Viewer | Broad Institute | https://software.broadinstitute.org/software/igv/; RRID:SCR_011793 | |
| Software, algorithm | Hi-C pipeline | n/a | https://github.com/danrobertson87/Barton_2021 | |
| Software, algorithm | ChIP-seq pipeline | n/a | https://github.com/danrobertson87/Barton_2021 | |
| Software, algorithm | HiGlass | Harvard, MIT | https://higlass.io/ | |
| Other | Protein G Dynabeads | Invitrogen | Cat# 10009D | |
| Other | Chelex-100 Resin | Bio-Rad | Cat# 1421253 | |
| Other | AMPure XP bead | Beckman | Cat# A63881 | |
| Other | Dynabeads MyOne Streptavidin C1 beads | Invitrogen | Cat# 65002 | |
| Other | Ibidi dishes μ-Slide 8-Well Glass | Ibidi | Cat# 80827 | |
| Other | DNA LoBind tubes | Eppendorf | Cat# 0030 108.051 | |
| Other | MiniSeq High output reagent kit (150-cycles) | Illumina | Cat# FC-420-1002 | |
| Other | Amicon 30 kDa columns | Merck | Cat# ufc903008 | |
| Other | NEXTflex-6 DNA Barcodes | PerkinElmer | Cat# 514101 | |

## Yeast strains and plasmids

Yeast strains used in this study were either *Saccharomyces cerevisiae* SK1 or W303 derivatives or *Schizosaccharomyces pombe*, all of which are listed in *Supplementary file 1*. Plasmids used in this study are listed in the Key resources table. Gene tags, gene deletions, and promoter replacements were introduced using standard PCR-based methods. Specific depletion of proteins (Cdc20, Smc3, Cdc6) during meiosis was achieved by placement of genes under the mitosis-specific *CLB2* or *SCC1* promoters (*Lee and Amon, 2003*). For early meiotic block/release experiments, strains carried *pCUP1-IME1 pCUP1-IME4* (*Berchowitz et al., 2013*); for prophase block/release experiments, strains carried *pGAL1-NDT80 pGPD1-GAL4(848).ER* (*Benjamin et al., 2003*; *Carlile and Amon, 2008*). For experiments undertaken in a prophase I arrest, the strains carried *ndt80Δ* (*Xu et al., 1995*), while for experiments in a metaphase I arrest cells carried *pCLB2-CDC20*. For experiments in an early meiotic arrest, strains carried *pCUP1-IME1 pCUP1-IME4* and were not induced for *pCUP1* expression. All anchor-away strains and their controls carried *RPL13A-2xFKBP12 fpr1Δ tor1-1* (*Haruki et al., 2008*) (kind gift from Andreas Hochwagen, NYU, NY). For centromeric GFP foci assays, strains carried *CEN5::tetOx224 pURA3-TetR-GFP* (*Tanaka et al., 2000*), and for arm GFP foci assays strains carried *lys2::tetOx240 pURA3-TetR-GFP* (*Brar et al., 2009*). For ChIP-qPCR, strains carried *REC8-3HA* (*Klein et al., 1999*), *SPO13-3FLAG, REC8-6HIS-3FLAG, SGO1-6HA*. To generate yeast strains carrying *SMC3* or *smc3-K112,113R*, *SMC3* was amplified from genomic DNA and cloned into YIplac128 to generate AMp1342. A plasmid with *smc3-K112,113R* (AMp1392) was generated by site-directed mutagenesis (using QuikChange XL Site-directed mutagenesis kit, Agilent). Both plasmids were integrated into the *LEU2* locus in the *smc3-md* strain background.

## Growth conditions

For asynchronous mitotic culture, cells were inoculated into YPDA (1% Bacto yeast extract, 2% Bacto peptone, 2% glucose, 0.3 mM adenine) and grown at 30°C with shaking at 250 rpm for ~15 hr. Cells

were diluted to $OD_{600}$ = 0.2 in YPDA and grown at room temperature for approximately 5 hr until $OD_{600}$ = 0.7–1.0. The $OD_{600}$ was measured, and the cells were harvested for the experiment. For testing the viability of anchor-away strains, YPDA agar plates contained 0.1 µM rapamycin.

For induction of meiosis, diploid yeast strains were thawed onto YPG agar (1% Bacto yeast extract, 2% Bacto peptone, 2.5% glycerol, 0.3 mM adenine, 2% agar) and incubated at 30°C overnight before transferring onto 4% YPDA agar (1% Bacto yeast extract, 2% Bacto peptone, 4% glucose, 0.3 mM adenine, 2% agar) and incubating at 30°C 8 hr or overnight. Cells were inoculated into YPDA and shaken at 30°C for 24 hr, diluted into BYTA (1% Bacto yeast extract, 2% Bacto tryptone, 1% potassium acetate, 50 mM potassium phthalate) at $OD_{600}$ = 0.2 and incubated at 30°C with shaking at 250 rpm for 14–16 hr. Cells were collected by centrifugation, washed twice in sterile $dH_2O$, resuspended in SPO (0.3% potassium acetate, pH 7) at $OD_{600}$ = 1.8–1.9, and incubated at 30°C with shaking at 250 rpm to induce meiosis (t = 0). For synchronous induction of S phase, to cells carrying *pCUP1-IME1 pCUP1-IME4* (*Berchowitz et al., 2013*), 25 µM $CuSO_4$ was added 2 hr after resuspension in SPO. For synchronous release from prophase, to cells carrying *pGAL1-NDT80 pGPD1-GAL4(848).ER* (*Benjamin et al., 2003*; *Carlile and Amon, 2008*) 1 µM β-estradiol was added 6 hr after resuspension in SPO.

For anchor-away experiments, all strains carried *tor1-1* and *fpr1Δ* mutations to prevent rapamycin toxicity in addition to Rpl13A-2xFKBP12. 1 µM rapamycin (5 mM rapamycin stock in dimethyl sulfoxide [DMSO]) or DMSO (as a control) was added to all cultures at the time of resuspension in SPO.

## Spore viability

Diploid strains were sporulated on agar (>48 hr) or liquid (24 hr) SPO media (with addition of appropriate drugs) at 30°C. Cells were treated with 20 µl 1 mg/ml zymolyase (AMS Biotechnology, Abingdon, UK) in 2 M sorbitol for 8 min, before diluting with 1 ml $dH_2O$. A minimum of 50 tetrads were dissected into individual spores on YPDA agar using a micromanipulator on a Nikon Eclipse 50i light microscope and viable colonies were scored ~48 hr later.

## DNA staining

100 µl meiotic cell culture was added to 400 µl of 100% EtOH and stored at 4°C. For DNA staining, the cells were pelleted and resuspended in 20 µl of 1 µg/ml DAPI in PBS (13.7 mM NaCl, 270 µM KCl, 1 mM $Na_2PO_4$, 176 µM $KH_2PO_4$) and stored at 4°C. For visualization and scoring, 3 µl of sample on a 1-mm-thick glass slide (Fisher Scientific) and an 18 × 18 mm cover slip (VWR, Radnor, PA) was imaged on a Zeiss Axioplan Imager Z2 fluorescence microscope with a 100× Plan ApoChromat NA 1.45 oil lens. For each condition, 200 cells were scored.

## Tubulin immunofluorescence

300 µl of meiotic cell culture was pelleted, resuspended in 500 µl of 3.7% formaldehyde solution (3.7% formaldehyde diluted in 0.1 M potassium phosphate buffer pH 6.4), and placed at 4°C overnight. Cells were pelleted and washed three times by resuspending in 1 ml of 0.1 M potassium phosphate buffer pH 6.4, then pellets were resuspended in 1.2 M sorbitol-citrate (1.2 M sorbitol, 0.1 M $K_2HPO_4$, 36 mM citric acid). The cells were then either stored at –20°C or the immunofluorescence protocol immediately continued. Fixed, washed cells were pelleted, resuspended in 226 µl of Digestion Solution (200 µl 1.2 M sorbitol-citrate, 20 µl glusulase [PerkinElmer, Waltham, MA], 6 µl 10 mg/ml zymolyase [AMS Biotechnology]), and incubated at 30°C for 2–3 hr until spheroplasts were observed by light microscopy. The digested cells were pelleted at 3000 rpm for 2 min, gently washed in 1.2 M sorbitol-citrate, resuspended in approximately 30 µl 1.2 M sorbitol-citrate before adhering 5 µl of cell suspension to each well of a multi-well slide (pretreated by addition of 5 µl of 0.1% polylysine for 5 min, washing in $dH_2O$ and air drying) for 10 min. The supernatant was aspirated, cell density confirmed by light microscopy, then the slide was incubated in 100% MeOH for 3 min followed by 10 s in 100% acetone before air drying. To each well, 5 µl primary rat anti-tubulin antibody (Bio-Rad AbD Serotec, Hercules, CA) at a 1:50 dilution in PBS/1% BSA was added and the slide incubated in a dark wet chamber at room temperature for 2 hr. Wells were washed individually five times in 5 µl PBS/BSA with aspiration between each wash. Secondary donkey anti-rat FITC antibody (Jackson ImmunoResearch, Ely, UK) was added at a 1:100 dilution in PBS/BSA (5 µl) and incubated for a further 2 hr before washing five times as above. DAPI-Mount (1 mg/ml *p*-phenylenediamine, 0.05 µg/ml DAPI, 40 mM $K_2HPO_4$, 10 mM $KH_2PO_4$, 150 mM NaCl, 0.1% $NaN_3$, 90% glycerol) was added to each well (3

µl), and the slide was covered with a 24 × 60 mm cover slip and sealed with nail varnish. Slides were then stored at –20°C before visualizing on a Zeiss Axioplan Imager Z2 fluorescence microscope with a 100× Plan ApoChromat NA 1.45 oil lens. Spindle morphology was scored in 200 cells per timepoint.

## Flow cytometry

For flow cytometry, 150 µl of meiotic culture was fixed in 70% EtOH and stored at 4°C. Pelleted cells were resuspended in 1 ml 50 mM Tris pH 7.5 and sonicated in a Bioruptor Twin (Diagenode, Liège, Belgium) on LOW for 30 s ON. Pellets were collected, resuspended in 475 µl of 50 mM Tris-HCl pH 7.5 with 25 µl 20 mg/ml RNase A (Amresco-VWR), and incubated at 37°C overnight. Cell pellets were washed in 1 ml 50 mM Tris pH 7.5, resuspended in 500 µl 50 mM Tris pH 7.5 with 10 µl 20 mg/ml Proteinase K (Invitrogen, Waltham, MA) and placed at 50°C for 2 hr. Collected cell pellets were washed in 1 ml 50 mM NaCitrate, resuspended in 500 µl 50 mM NaCitrate with 9.17 µl 1 mg/ml propidium iodide (Sigma-Aldrich, Darmstadt, Germany), and sonicated on LOW 30 s ON, 30 s OFF 10 times. Samples were measured on a Becton Dickinson FACSCalibur with CellQuest Pro program or an Attune NxT flow cytometer (n = 20,000 cells per sample) and analyzed using FlowJo V10. Signal intensity is shown on a linear scale.

## Visualization of GFP-labeled chromosomes

100 µl of culture was added to 10 µl of 37% formaldehyde and incubated for 8–9 min at room temperature before pelleting. The supernatant was aspirated, 1 ml of 80% EtOH added, and tubes briefly vortexed. Cells were collected by short (~15 s) centrifugation, all traces of 80% EtOH removed by pipetting, the pellet was resuspended in 20 µl of 1 µg/ml DAPI in PBS, and stored at 4°C. For visualization, 3 µl of sample was transferred to a glass slide, a coverslip added, pressure applied to flatten the cells before viewing under a Zeiss Axioplan Imager Z2 fluorescence microscope with a 100× Plan ApoChromat NA 1.45 oil lens. All unbudded cells were scored for the presence of either one or two GFP dots (n = 100). All experiments were done in triplicate.

## Live-cell imaging

8-Well Glass Bottom µ-Slide Ibidi dishes (Ibidi) were prepared by spreading 45 µl 5 mg/ml concanavalin A (dissolved in 50 mM $CaCl_2$, 50 mM $MnCl_2$) evenly in the bottom of each well and incubating at 32°C for 15 min. Excess concanavalin A was removed by aspiration, and each well washed three times in 500 µl $dH_2O$. Around 2.30 hr after meiotic cultures were resuspended in SPO ($OD_{600}$ = 2.5) as described above, 3 ml of cell culture was harvested by centrifugation (3000 rpm, 3 min) to concentrate, resuspended in 300 µl of their supernatant (preconditioned), and added to the wells of the Ibidi dish. The dish was incubated at 30°C for 20 min, the excess SPO culture aspirated, and wells were washed twice in 500 µl $dH_2O$ and once in 500 µl of preconditioned SPO media, before adding 400 µl preconditioned SPO media to each well. Cells were visualized with a Zeiss Axio Observer Z1 (Zeiss UK, Cambridge, UK) with a Prior motorized stage equipped with a Hamamatsu Flash 4 sCMOS camera and Zen 2.3 acquisition software on a Linux computer. Images were acquired every 15 min for 12 hr, with eight Z-stacks of 0.8 µM for FITC and Tomato channels and a single stack for Brightfield. For CEN5-GFP dots, FITC channel imaging conditions were binning 2 × 2, 5% transmitted light, and 0.15 s exposure. For imaging of Spc42-tdTomato and Pds1-tdTomato, the red channel imaging conditions were binning 2 × 2, 5% transmitted light, and 0.2 s exposure. The images were analyzed using the ImageJ software version 2.0.0-rc-43/1.51g (National Institutes of Health, Bethesda, MD). For each experiment, all strains were imaged simultaneously in multi-well Ibidi dishes and experiments were performed at least twice, scoring at least 100 cells per strain. A representative experiment is shown.

## Western blotting

For mitotic protein extracts, 10 ml of YPDA cell culture $OD_{600}$ = 0.6–1 was collected, and for meiotic protein extracts 5 ml of SPO cell culture $OD_{600}$ = 1.8 was collected. Cells were pelleted by centrifugation, resuspended in 5 ml 5% trichloroacetic acid, and incubated on ice for a minimum of 10 min, before pelleting and transferring to a 2 ml fast-prep tube (MP Biomedicals). The fast-prep tube was centrifuged, the supernatant removed and snap-frozen in liquid nitrogen before storing at –80°C. Thawed samples were resuspended in 1 ml acetone by vortexing, cells pelleted by centrifugation, and the acetone removed. Samples were air-dried until the pellet was dry (>4 hr) and resuspended

in 100 µl of protein breakage buffer (10 mM Tris-HCl pH 7.5, 1 mM EDTA pH 7.5, 2.75 mM DTT, 1×
Roche EDTA-free protease inhibitors) and one volume of glass beads (0.5 mm zirconia/silica glass
beads, Biospec Products, Bartlesville, OK) added. The cells were disrupted in a Fastprep Bio-Pulverizer
FP120 at 6.5 speed for three cycles of 45 s, placed on ice, and 50 µl of 3× SDS sample buffer (187 mM
Tris-HCl pH 6.8, 6% β-mercaptoethanol, 30% glycerol, 9% SDS, 0.05% bromophenol blue) was added
to the lysate before immediately heating at 95°C for 5 min, cooled and centrifuged before loading
onto SDS-PAGE gels of the appropriate concentration (8–10%). PAGE was carried out using a Bio-Rad
Mini Trans-Blot System (Bio-Rad) or a Biometra V17.15 System (Biometra, Milano, Italy) in SDS running
buffer (25 mM Tris, 190 mM glycine, 0.01% SDS). SDS-PAGE gels were transferred onto nitrocellulose
membrane (0.45 µM, Amersham-GE Healthcare, Amersham, UK) in transfer buffer (25 mM Tris, 1.5%
glycine, 0.02% SDS, 10% MeOH) in either a Bio-Rad Mini Trans-Blot system or a semi-dry Amersham
TE70 transfer unit. Membranes were blocked in 5% milk in PBS with 0.05% Tween20 (PBST) for at least
1 hr at room temperature before incubating in primary antibody in 2% milk/PBST overnight at 4°C.
Membranes were washed in PBST three times for 15 min, incubated in secondary antibody in 2% milk/
PBST for 1 hr at room temperature, and washed in PBST three times. HRP-conjugated antibodies were
detected with an ECL SuperSignal West Pico chemiluminescence kit (Thermo Scientific, Waltham,
MA) ECL SuperSignal West Pico, supplemented with SuperSignal West Femto chemiluminescence
kit (Thermo Scientific) for weaker signals. Membranes were exposed to CP-BU X-ray film (Agfa
Healthcare, Mortsel, Belgium), developed using a Konica-Minolta SRX-101A developer, or images
were directly acquired with a ChemiDoc MP Imaging System (Bio-Rad). Primary antibodies used were
mouse anti-HA 12CA5 (Roche; 1:1000), mouse anti-HA11 (BioLegend, San Diego, CA; 1:1000), mouse
anti-FLAG M2 (Sigma-Aldrich; 1:1000), rabbit anti-Pgk1 (Marston lab stock; 1:10,000), rabbit anti-Kar2
(Marston lab stock; 1:10,000), rabbit anti-Smc3-K112,113Ac (custom-made, GenScript, Piscataway,
NJ; 1:1000), rabbit anti-Rec8 (Marston lab stock; 1:15,000), and rabbit anti-Smc3 (Marston lab stock;
1:5000). Secondary antibodies used were sheep anti-mouse HRP (GE Healthcare, Chicago, IL; 1:5000)
and donkey anti-rabbit (GE Healthcare; 1:5000).

## Chromatin immunoprecipitation (ChIP)-qPCR

Meiotic culture (45 ml) was mixed with 5 ml of fixing solution (1.5 ml 37% formaldehyde in 3.5 ml
diluent [143 mM NaCl, 1.43 mM EDTA, 71.43 mM HEPES-KOH pH 7.5]), and gently rocked at room
temperature for 2 hr. Cells were pelleted and washed twice in 10 ml of ice-cold TBS (20 mM Tris-HCl pH
7.5, 150 mM NaCl), then once in 10 ml ice-cold 1xFA lysis buffer (50 mM HEPES-KOH pH 7.5, 150 mM
NaCl, 1 mM EDTA, 1% Triton X-100, 0.1% Na deoxycholate) with 0.1% SDS. Pellets were collected
into 2 ml fast-prep tubes (MP Biomedicals, Santa Ana, CA), snap-frozen in liquid nitrogen, and stored
at –80°C. Cell pellets were thawed on ice, and the pellet resuspended in 300 µl 1xFA lysis buffer
with 0.5% SDS, 1× Roche EDTA-free protease inhibitors, 1 mM PMSF, and one scoop of glass beads
(0.5 mM zirconia/silica glass beads, Biospec Products) was added. Cells were disrupted in a Fastprep
Bio-Pulverizer FP120 at 6.5 speed for two cycles of 30 s, with a 10 min waiting period on ice between.
The tube was punctured, the cell lysate and debris were collected by centrifugation, transferred to
a new tube, and then pelleted by centrifugation at 13,000 rpm for 15 min at 4°C. The cell pellet was
resuspended in 1 ml 1xFA lysis buffer with 0.1% SDS, 1× Roche EDTA-free protease inhibitors and
1 mM PMSF, and re-centrifuged. The supernatant was removed, the pellet resuspended in 300 µl 1xFA
lysis buffer with 0.1% SDS, 1× Roche EDTA-free protease inhibitors and 1 mM PMSF, then sonicated
at 4°C in a Bioruptor Twin sonicating device (Diagenode) on HIGH 30 s ON, 30 s OFF for 30 min total.
The cell debris was pelleted at 13,000 rpm for 15 min at 4°C, and the supernatant transferred into a
new 1.5 ml tube containing 1 ml of 1xFA lysis buffer with 0.1% SDS, 1× Roche EDTA-free protease
inhibitors and 1 mM PMSF. The chromatin lysate was re-centrifuged, and the supernatant transferred
into a new tube. For the INPUT sample, 10 µl of the supernatant was frozen at –20°C overnight. For
the IP, 15 µl of 20 mg/ml Protein G Dynabeads (Invitrogen) per sample were pre-washed four times in
1 ml 1xFA lysis buffer with 0.1% SDS, 1× Roche EDTA-free protease inhibitors and 1 mM PMSF, then
resuspended in 100 µl/sample 1xFA lysis buffer with 0.1% SDS, 1× Roche EDTA-free protease inhib-
itors and 1 mM PMSF. To 1 ml of cell supernatant, 100 µl of Protein G Dynabeads was added, along
with the appropriate antibody (mouse HA 12CA5, 7.5 µl 0.4 mg/ml, Roche, Basel, Switzerland; mouse
M2 FLAG, 5 µl 1 mg/ml, Sigma), before incubating at 4°C on a rotating wheel at 14 rpm for 15–18 hr.
Following incubation, the beads were collected on a magnet, and the supernatant discarded. The

beads were washed 5 min/wash consecutively in ChIP wash buffer 1 (1xFA lysis buffer, 0.1% SDS, 275 mM NaCl), ChIP wash buffer 2 (1xFA lysis buffer, 0.1% SDS, 500 mM NaCl), ChIP wash buffer 3 (10 mM Tris-HCl pH 8, 250 mM LiCl, 1 mM EDTA, 0.5% NP-40, 0.5% Na deoxycholate), and ChIP wash buffer 4 (10 mM Tris-HCl pH 8, 1 mM EDTA), with 30 s on a small magnet in-between each wash. After the final wash, all of the supernatant was removed from the beads.

Chelex-100 Resin (Bio-Rad) was resuspended at 0.1 g/ml in HyClone Water (Hypure Molecular Biology Grade Water, GE Healthcare), and 100 µl added to both the thawed and vortexed INPUT samples and the IP samples. The samples were boiled at 100°C for 10 min, cooled on ice, then briefly centrifuged at 2000 rpm for 1 min. To each tube, 2.5 µl 10 mg/ml Proteinase K (Invitrogen) was added, the samples vortexed, and then incubated at 55°C for 30 min. The samples were again boiled at 100°C for 10 min, cooled on ice, then briefly centrifuged at 2000 rpm for 1 min. Approximately 120 µl of supernatant was transferred into a new tube and frozen at –20°C. For qPCR, either SYBR GreenER (Invitrogen; *Figure 8—figure supplement 2A,B*) or Luna (NEB, Ipswich, MA; *Figure 8—figure supplement 2D*) mastermix was used. Input DNA was diluted 1:500 or 1:300 and the ChIP DNA was diluted 1:10 or 1: 6 in HyClone Water for SYBR GreenER or Luna, respectively. For each biological replicate, qPCR reactions were carried out in technical triplicate on a LightCycler 480 Roche machine with 40 cycles for SYBR GreenER and 45 cycles for NEB Luna. The threshold cycle (Ct) values were computed by the LightCycler 480 Roche software using the second derivative maximum algorithm. The geometric mean of technical replicate Ct values was determined and delta Ct was calculated according to the following formula: $\Delta Ct = Ct_{(ChIP)} (Ct_{(Input)} \log_{(primer\ efficiency)}(\text{Input dilution factor}))$. Enrichment values are given as ChIP/Input = (primer efficiency)$^{(-\Delta Ct)}$. Primers used are given in *Appendix 1—table 1*.

## ChIP-sequencing

For calibrated ChIP-seq (Smc3), we used the method of *Hu et al., 2015*, as modified by *Galander et al., 2019b*. For each condition, 200 ml of *S. cerevisiae* meiotic culture was grown and fixed as for ChIP-qPCR and frozen in four pellets (each from 50 ml culture). Each pellet was mixed with a fixed pellet from 100 ml *S. pombe* culture prepared as follows. Wild-type *S. pombe* (strain spAM29) was inoculated into YES media (0.5% yeast extract, 3% glucose, 2% agar 225 mg/l each of adenine, histidine, leucin, uracil, and lysine hydrochloride), grown for 16 hr at 30°C before diluting to $OD_{600} = 0.07$–0.1 and growing for approximately 4–6 hr at 30°C until the $OD_{600} = 0.4$. Cell culture (1 l) was mixed with 100 ml of fixing solution (33.3 ml 37% formaldehyde in 66.6 ml diluent) and incubated at room temperature at 90 rpm for 2 hr. Cells were harvested, washed twice with TBS, and completely resuspended in 1xFA lysis/0.1% SDS. An equal amount of cells, equivalent to 100 ml of culture, was evenly distributed into Fastprep tubes and snap-frozen. Thawed *S. cerevisiae* and *S. pombe* cell pellets were combined on ice in 400 µl 1xFA lysis buffer with 0.5% SDS, 1× Roche EDTA-free protease inhibitors and 1 mM PMSF. Cells were processed as for ChIP-qPCR except four cycles of 60 s on Fastprep Bio-Pulverizer FP120, and two rounds of 30 cycles 30 s ON-30 s OFF of sonication with Bioruptor Plus (Diagenode) set on HIGH were performed. Before setting up the IP, the four samples of each condition were pooled together and then split into a 10 µl INPUT and 4 × 1 ml IPs, to whom 15 µl of Protein G Dynabeads (Invitrogen) and 10 µl of rabbit anti-Smc3 (Marston lab stock) were added. After overnight incubation, washes were performed as for ChIP-qPCR, and after the final wash the beads for each condition were pooled in 200 µl TES (50 mM Tris-HCl pH 7.5, 10 mM EDTA, 1% SDS), the sample eluted at 65°C for 20 min, and transferred to a new tube. The beads were washed for 15 min at room temperature in 200 µl TE (10 mM Tris-HCl pH 7.5, 1 mM EDTA pH 7.5), which was then combined with the corresponding eluate. 40 µl 10 mg/ml Proteinase K was added, and the samples were incubated at 42°C for 1 hr then at 65°C for 16 hr. Samples were cleaned up using a Promega Wizard Kit, eluted in 35 µl HyClone Water and frozen at –20°C.

Non-calibrated ChIP-seq (Rec8) samples were processed with the same steps and conditions as the calibrated ones, but without adding *S. pombe* cell pellets.

ChIP-sequencing libraries were prepared using NEXTflex-6 DNA Barcodes (PerkinElmer) in DNA LoBind tubes (Eppendorf, Hamburg, Germany) using HyClone Water. Briefly, 2 ng of INPUT or IP DNA was taken, and blunt and phosphorylated ends were generated using the Quick Blunting kit (NEB), before removal of DNA fragments under 100 bp using AMPure XP beads (Beckman Coulter), followed by addition of dA tails by Klenow fragment (*exo-*) enzyme (NEB). Suitable NEXTflex-6 barcodes were

ligated using Quick Ligation kit (NEB) before consecutive selection of DNA fragments > 100 bp and then >150–200 bp using AMPure XP beads. Libraries were amplified by PCR using NextFlex PCR primers (primer 1–5'-AATGATACGGCGACCACCGAGATCTACAC; primer 2–5'-CAAGCAGAAGAC GGCATACGAGAT) and Phusion High-Fidelity DNA Polymerase (NEB) before three further rounds of AMPure XP purification were performed to collect fragments 150–300 bp in size. Library quality was assessed on a Bioanalyzer (Agilent) using the 2100 Bioanalyzer High Sensitivity DNA kit (Agilent, Santa Clara, CA) and DNA was quantified by Qubit (Thermo Scientific), before preparation and denaturation of a 1 nM pooled library sample where INPUTs and IPs were mixed in 15–85% ratio, followed by sequencing in house on an Illumina MiniSeq instrument (Illumina, San Diego, CA) with MiniSeq High output reagent kit (150-cycles) (Illumina). Calculation of occupancy ratio (OR) and data analysis was performed as described by *Hu et al., 2015*. Briefly, reads were mapped to both the *S. pombe* calibration and *S. cerevisiae* SK1 experimental genomes and the number of reads mapping to each genome was determined. OR was then determined using the formula Wc * IPx/Wx * IPc, where W = input, IP = ChIP, c = calibration genome (*S. pombe*), and x = experimental genome (*S. cerevisiae* SK1). The number of reads at each position was normalized using the OR and visualized using the Integrated Genome Viewer (IGV, Broad Institute, Cambridge, MA). Mean calibrated ChIP-seq read plots (pileup) at centromeres, pericentromeric borders, and arm sites were generated using the Bioconductor SeqPlots package. Reads were binned at 50 bp windows around midpoint of centromeres/pericentromeric borders/arms with 3 kb flanks at either side. Pericentromeric borders were oriented so that their position relative to the centromere was the same. To obtain genomic coordinates of borders in SK1, border cohesin peaks corresponding to those defined in w303 (*Paldi et al., 2020*) were identified from wild-type Smc3 ChIP-seq tracts visualized in IGV (either homologous sequence or, where the sequence was divergent, the position between the same gene pair). For the plots of cohesin enrichment on chromosome arms, the coordinates were chosen as for pericentromere borders; when W303 coordinates did not correspond to a cohesin peak in SK1, the next peak (distal from the centromere) was chosen. Centromere, pericentromere border, and arm peak coordinates are listed in *Appendix 1—table 2*.

## Hi-C

The Hi-C protocol was adapted from *Paldi et al., 2020* and *Schalbetter et al., 2019*. Briefly, 50 ml of synchronized meiotic culture at $OD_{600}$ ~ 2 was fixed for 20 min in 3% formaldehyde at room temperature with 90 rpm shaking. The reaction was quenched by adding glycine to 0.35 M and incubating for 5 min. Cells were harvested and washed once in 40 ml of cold water, resuspended in 5 ml 1× NEB Buffer 2 (NEB) and drop frozen in liquid nitrogen. Frozen pellets were ground in a chilled pestle and mortar on dry ice for 20 min with addition of liquid nitrogen every 3 min. About 0.5 g of crushed 4N cell pellets or 1 g of 2N cell pellets (adjusted to the measured culture $OD_{600}$) were then thawed, washed with 1× NEB Buffer 3.1 (NEB) and digested with 2 U/μl of DpnII restriction nuclease (NEB) at 37°C overnight. Digested DNA ends were filled-in and biotinylated with a nucleotide mix containing 0.4 mM biotin-14-dCTP (Invitrogen) instead of dCTP and 5 U/μl Klenow fragment Dpol I (NEB) at 37°C for 2 hr. The reaction was stopped by incubating at 65°C for 20 min with 1.5% SDS. Proximity ligation was then carried out on diluted samples (25 ml) with 0.024 U/μl of T4 DNA ligase (Invitrogen) at 16°C for 8 hr, after which the ligase was inactivated with 5.3 mM EDTA pH8.0. DNA was decrosslinked overnight at 65°C with 70 μg/μl proteinase K (Invitrogen), topped up to 140 μg/μl for the final 2 hr, and then DNA was extracted with 20 ml phenol:chloroform:isoamyl alcohol 25:24:1 (Sigma), precipitated with ethanol, and resuspended in 22.5 ml TE buffer pH 8.0. To concentrate and desalt DNA, samples were filtered through Amicon 30 kDa columns (Merck). DNA was then subjected to a second round of phenol extraction and ethanol purification, before being resuspended in 20 μl of TE and treated first with 1 mg/ml RNAse A (Amresco) for 1 hr at 37°C, next with 3U of T4 DNA Polymerase (NEB) for each μg of DNA and 5 mM dNTPs for 4 hr at 20°C to remove biotin from unligated ends. DNA was fragmented with two rounds of 30 cycles 30 s ON–30 s OFF of sonication with Bioruptor Plus (Diagenode) set on HIGH and purified with QIAGEN MinElute kit (QIAGEN, Hilden, Germany). After fragmentation, DNA ends were repaired by treatment with 3U T4 DNA polymerase (NEB), 50U T4 Polynucleotide Kinase (NEB), and 25 U Klenow fragment DPol I (NEB) for 30 min at 20°C. DNA was once again purified with a QIAGEN MinElute kit and dA-tailed with 11.25U Klenow fragment (exo-) enzyme (NEB). DNA fragments were then selected for size (200–300 bp) with two

subsequent rounds of AMPure XP bead (Beckman, Pasadena, CA) treatment and for the presence of the Biotin marker by Dynabeads MyOne Streptavidin C1 beads (Invitrogen) pull-down. Finally, NEXTflex-6 barcoded adapters (PerkinElmer) were ligated using 1 µl Quick T4 DNA Ligase (NEB) and DNA fragments were amplified by PCR with NEXTFlex primers (primer 1–5'-AATGATACGGCGACCA CCGAGATCTACAC; primer 2–5'-CAAGCAGAAGACGGCATACGAGAT) and Phusion High-Fidelity DNA Polymerase (NEB) and further purified with AMPure XP beads to eliminate unligated primers and adapters. Libraries were sequenced with 42 bp paired-end reads on an Illumina NextSeq500 (EMBL Core Genomics Facility, Heidelberg, Germany). Hi-C read numbers for each library are listed in *Appendix 1—table 3*.

For Hi-C data analysis, FastQ reads were aligned to *S. cerevisiae* SK1 reference genome using HiC-Pro v2.11.4 bowtie2 v2.3.4.1 (`--very-sensitive -L 30 --score-min L,–0.6,–0.2 --end-to-end --reorder`), removing singleton, multi hit, duplicated, and MAPQ < 10 reads. Read pairs were assigned to restriction fragment (*Dpn*II) and invalid pairs filtered out. Valid interaction pairs were converted into the .cool contact matrix format using the cooler library, and matrices balanced using Iterative correction down to 1 kb resolution. Multiresolution cool files were uploaded onto a local HiGlass server. To generate pileups at centromeres/pericentromeric borders, the cooltools library was used and cool matrices were binned at 1 kb resolutions. Plots were created around the midpoint of centromeres/pericentromeric borders at the same coordinates used for ChIP-seq plots (listed in *Appendix 1—table 2*), with 25, 50, and 100 kb flanks on each side, showing the $\log_{10}$ mean interaction frequency using a color map similar to HiGlass 'fall'. All centromere annotations were duplicated in both the forward/reverse strand orientations to create an average image, which is mirror symmetrical. Pileups at pericentromeric borders were oriented where positions are identical in relation to the centromere. The ratio pileups between samples were created in a similar fashion plotting the $\log_2$ difference between samples in the 'coolwarm' color map, for example, A/B; red signifying increased contacts in A relative to B and blue decreased contacts in B relative to A. Scripts for visualization are available at https://github.com/danrobertson87/Barton_2021; *Barton, 2022* (copy archived at swh:1:rev:a457e30eb49b9a587605c91ed5d878735d6f9e51). Cooler 'show' was also used to generate individual plots for each chromosome. Contact probability *P*(*s*) and its derivative slope plots were generated using the cooltools library. White stripes on plots represent regions where data was lost stochastically during mapping due to stringency settings filtering out reads. Hi-C matrices were aligned to ChIP-seq tracks on HiGlass.

## Source data

Original scans of blots and annotated versions are provided as source data.

## Acknowledgements

We are grateful to Andreas Hochwagen for yeast strains, Jean-Paul Javerzat, Lori Koch, Flora Paldi, and Gerard Pieper for comments on the manuscript, and to Akira Shinohara for helpful discussions. We thank Flora Paldi for advice on Hi-C sample preparation, and Weronika Borek, Julie Blyth, and Karolina Lesniewska for technical assistance with antibody generation. We gratefully acknowledge the EMBL Genecore for DNA sequencing, the Wellcome Centre for Cell Biology Core Bioinformatics Facility for sequencing analysis, and the Wellcome Centre Optical Imaging Laboratory (COIL) and Dave Kelly for microscopy and flow cytometry support. This work was funded through a Wellcome Senior Research Fellowship [107827] (ALM, LFM, REB), a Wellcome Investigator award [220780] (ALM, LFM), BBSRC grant [BB/S018018/1] (ALM, LFM), a Wellcome PhD studentship [102316] (REB), and core funding for the Wellcome Centre for Cell Biology [203149] (REB, LFM, DR, ALM).

## Additional information

### Competing interests

Adèle L Marston: Reviewing editor, *eLife*. The other authors declare that no competing interests exist.

## Funding

| Funder | Grant reference number | Author |
|---|---|---|
| Wellcome Trust | 107827 | Rachael E Barton<br>Lucia F Massari<br>Adèle L Marston |
| Wellcome Trust | 220780 | Lucia F Massari<br>Adèle L Marston |
| Wellcome Trust | 102316 | Rachael E Barton |
| Wellcome Trust | 203149 | Rachael E Barton<br>Lucia F Massari<br>Daniel Robertson<br>Adèle L Marston |
| Biotechnology and Biological Sciences Research Council | BB/S018018/1 | Rachael E Barton<br>Lucia F Massari<br>Adèle L Marston |

The funders had no role in study design, data collection and interpretation, or the decision to submit the work for publication.

## Author contributions

Rachael E Barton, Lucia F Massari, Conceptualization, Investigation, Methodology, Visualization, Writing - review and editing; Daniel Robertson, Formal analysis, Investigation, Methodology, Software, Visualization, Writing - review and editing; Adèle L Marston, Conceptualization, Funding acquisition, Investigation, Methodology, Supervision, Visualization, Writing - original draft, Writing - review and editing

## Author ORCIDs

Rachael E Barton http://orcid.org/0000-0001-8553-9029
Lucia F Massari http://orcid.org/0000-0002-6456-9699
Daniel Robertson http://orcid.org/0000-0003-3894-7200
Adèle L Marston http://orcid.org/0000-0002-3596-9407

## Decision letter and Author response

Decision letter https://doi.org/10.7554/eLife.74447.sa1
Author response https://doi.org/10.7554/eLife.74447.sa2

---

# Additional files

## Supplementary files
• Supplementary file 1. *Saccharomyces cerevisiae* (SK1 or W303) and *Schizosaccharomyces pombe* strains used in this study.

• Transparent reporting form

## Data availability

The data discussed in this publication have been deposited in NCBI's Gene Expression Omnibus (Edgar et al., 2002) and are accessible through GEO Series accession number GSE185021.

The following dataset was generated:

| Author(s) | Year | Dataset title | Dataset URL | Database and Identifier |
|---|---|---|---|---|
| Barton R E, Massari L F, Robertson D, Marston A L | 2022 | Eco1-dependent cohesin acetylation anchors chromatin loops and cohesion to define functional meiotic chromosome domains | https://www.ncbi.nlm.nih.gov/geo/query/acc.cgi?acc=GSE185021 | NCBI Gene Expression Omnibus, GSE185021 |

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

# Appendix 1

**Appendix 1—table 1.** List of qPCR primers used in this study.

| Chr | Location | Distance from centromere | Primer pair | Sequence | Figure |
|---|---|---|---|---|---|
| IV | Arm | −95 kb | 782 | AGATGAAACTCAGGCTACCA | |
| | | | 783 | TGCAACATCGTTAGTTCTTG | |
| IV | Pericentromere | −9.5 kb | 1319 | ATGATTCAATGGATTTAGCC | |
| | | | 1320 | GTCAGTCTTATGCTGTTCCC | |
| IV | Centromere | +150 bp | 794 | CCGAGGCTTTCATAGCTTA | *Figure 8—figure supplement 2A and B* |
| | | | 795 | ACCGGAAGGAAGAATAAGAA | |
| III | Centromere | −42 bp | 8196 | ATAAACCAAACCCTTCCCCTTC | |
| | | | 8197 | CCATATTGTTTGGCGCTGAT | |
| IV | Arm | −95 kb | 8175 | GCTACCACCAATAACACAGTTGAG | |
| | | | 8176 | GTACCTTCCCTGATAATCCGTCT | |
| IV | Centromere | +51 bp | 8172 | GCCGAGGCTTTCATAGCTTA | |
| | | | 8173 | GACGATAAAACCGGAAGGAAG | |
| XII | Centromere | −45 bp | 8206 | GGTTTGTAGACAACCAAACTGGTG | *Figure 8—figure supplement 2E* |
| | | | 8207 | ACTCTTTACGCGGGTGTGTACT | |

**Appendix 1—table 2.** SK1 genome coordinates used to generate ChIP-seq and Hi-C pileup plots.

| Chr | Left arm peak | Left border | CEN | Right border | Right arm peak |
|---|---|---|---|---|---|
| I | 143,338 | 156,847 | 160,506–160,623 | 170,170 | 174,656 |
| II | 209,535 | 226,877 | 229,855–229,971 | 235,956 | 269,863 |
| III | 56,131 | 106,064 | 119,146–119,262 | 134,944 | 148,849 |
| IV | 412,039 | 452,984 | 461,938–462,053 | 468,978 | 191,167 |
| V | 143,196 | 150,521 | 156,659–156,776 | 170,642 | 194,898 |
| VI | 141,526 | 145,685 | 150,233–150,350 | 156,148 | 174,078 |
| VII | 480,232 | 500,759 | 508,233–508,352 | 515,930 | 563,982 |
| VIII | 72,321 | 95,832 | 100,974–101,091 | 105,373 | 114,788 |
| IX | 320,860 | 354,853 | 360,299–360,308 | 370,438 | 381,917 |
| X | 423,353 | 445,538 | 450,666–450,783 | 455,253 | 468,108 |
| XI | 417,329 | 438,536 | 445,725–445,841 | 452,317 | 477,072 |
| XIV | 132,607 | 137,781 | 154,749–154,867 | 165,549 | 185,746 |
| XIII | 249,427 | 254,063 | 266,751–266,870 | 278,483 | 293,988 |
| XIV | 592,647 | 625,567 | 637,885–638,002 | 656,081 | 660,807 |
| XV | 310,053 | 322,437 | 327,363–327,480 | 335,598 | 373,770 |
| XVI | 537,065 | 554,046 | 559,567–559,683 | 564,164 | 584,849 |

**Appendix 1—table 3.** Hi-C libraries generated in this study.

| Library name | Relevant genotype and stage | Total R1/R2 aligned reads (M) | Valid unique Hi-C pairs (M) |
|---|---|---|---|
| HiC1_28719_wt | *RPL13A-2xFKBP12 fpr1Δ tor1-1 ndt80Δ* prophase | 97,377,167/93,678,652 | 17,763,563 |
| *HiC1_28720_eco1-aa* | *RPL13A-2xFKBP12 fpr1Δ tor1-1* *ndt80Δ ECO1-FRB-GFP* prophase | 103,777,687/100,044,807 | 15,759,219 |
| HiC1_29750_wpl1 | *RPL13A-2xFKBP12 fpr1Δ tor1-1 ndt80Δ rad61Δ* prophase | 77,203,756/74,233,462 | 13,525,545 |
| HiC1_29781_eco1-aa_wpl1 | *RPL13A-2xFKBP12 fpr1Δ tor1-1 ndt80Δ ECO1-FRB-GFP rad61Δ* prophase | 95,353,060/92,175,659 | 11,292,250 |
| HiC2_11633_wt | *ndt80Δ* prophase | 175,716,902/172,151,440 | 18,256,545 |
| HiC2_28841_clb5_clb6 | *ndt80Δ clb5Δ clb6Δ* prophase | 160,012,048/156,826,193 | 20,128,179 |
| HiC3_28719_wt | *RPL13A-2xFKBP12 fpr1Δ tor1-1 ndt80Δ* prophase | 105,178,126/103,054,489 | 13,127,333 |
| HiC3_28720_eco1-aa | *RPL13A-2xFKBP12 fpr1Δ tor1-1 ndt80Δ* *ECO1-FRB-GFP* prophase | 141,270,515/138,743,876 | 18,613,322 |
| HiC3_31003_cdc6 | *RPL13A-2xFKBP12 fpr1Δ tor1-1 ndt80Δ cdc6-md* prophase | 108,810,021/106,207,643 | 16,583,830 |
| HiC3_20991_eco1-aa | *RPL13A-2xFKBP12 fpr1Δ tor1-1 ndt80Δ* *ECO1-FRB-GFP cdc6-md* prophase | 101,646,217/99,245,960 | 15,733,530 |
| HiC4_12145_wt | *pCUP1::IME1 pCUP1::IME4* meiotic G1 | 201,424,826/198,072,796 | 24,576,849 |
| HiC4_26946_clb5clb6 | *pCUP1::IME1 pCUP1::IME4 clb5Δ clb6Δ* meiotic G1 | 240,834,552/235,969,577 | 27,737,075 |

