## [Editor Report]

With a large quantity and high quality of data set, this paper elegantly shows the role of Eco1-dependent Smc3 acetylation plays a role in the establishment of chromatin boundary during yeast meiosis, which is independent of meiotic DNA replication. Eco1-directed boundary formation, which is not counteracted by Wapl, is critical for prophase I exit and sister chromatid separation in meiosis II. In addition, Eco1 antagonizes Wapl-mediated cohesin removal at centromeres for proper sister cohesion and mono-orientation of sister kinetochores in meiosis I.

---

## [Decision Letter]

**Decision letter after peer review:**

Thank you for submitting your article "Eco1-dependent cohesin acetylation anchors chromatin loops and cohesion to define functional meiotic chromosome domains" for consideration by *eLife*. Your article has been reviewed by 3 peer reviewers, including Akira Shinohara as Reviewing Editor and Reviewer #1, and the evaluation has been overseen by a Reviewing Editor and Jessica Tyler as the Senior Editor. The following individual involved in review of your submission has agreed to reveal their identity: Dean Dawson (Reviewer #2).

Thanks for giving us an opportunity to review your paper on "Eco1-dependent cohesin acetylation anchors chromatin loops and cohesion to define functional meiotic chromosome domains". As mentioned above, three reviewers as well as a senior editor, discussed your results, and all acknowledge the importance of this paper and recommend possible publication in *eLife*. However, we agree that additional experiments are required for revision of the paper, which described in detail below.

Essential revisions:

1. HiC analysis at 0 h in clb5, clb6: The authors nicely showed that the clb5 clb6 double mutant (and CDC6mn), which is defective in meiotic DNA replication, shows Eco1-dependent acetylation of Smc3 (Figure 1E and S1E). This is clearly difference from mitotic cells, which show replication-dependent Smc3 acetylation. The key finding in the paper would be on replication-independent border formation (Figure 5). One concern is whether this is indeed promoted by Smc3 acetylation or comes from chromosome memory of the boundary from previous mitotic cells. To distinguish these possibilities, the authors need to analyze boundary profiles in the clb5 clb6 double mutant (and wild type) at 0 h and compared to those in prophase I (Figure 5).

2. Live-imaging of sister centromeres during meiosis I: The role of Eco1-dependent Smc3 acetylation in promoting mono-polar kinetochore formation is one of the key findings (Figure 6) since previous studies showed mono-polar orientation of sister kinetochores during meiosis I does not depend on Rec8-cohesin in budding yeast. The authors only analyzed the results of chromosome segregation after anaphase I of the spo11 eco1-aa mutant. It would be helpful if the authors could see bipolar sister kinetochores on meiotic spindles during meiosis I. One concern is on precocious sister separation in late prophase I may cause similar defect in the assay (Figure 6C and D). The authors can check the status of sister centromeres in late prophase I, metaphase I and anaphase I in the spo11 eco1-aa mutant by live-imaging of cells with CenV-GFP used in Figures 2 and 6.

3. Discussion on functionality of Eco1-FRB-GFP: Since the FRB-GFP tag seems to affect the functionality of Eco1 protein in meiosis, it would be nice to carry out a complemental approach such as auxin-degron. However, there might be similar problem(s) for any tagging, even for N-terminal tagging. Thus, it is recommended that, by text rewriting, the authors describe more clearly about the limitation of the tag to know the Eco1 function in vivo and soften some conclusions in the text.

4. Figure 7H and a model: In this model, the authors suggest the model on a cohesin-mediated loop-extrusion and sister chromatid cohesion. As, in the paper, the authors showed two distinct cohesin-mediated structures on centromeres and chromosome arms (and pericentromeres). It would be helpful to readers if the authors provide models on how cohesin- and its acetylation-mediated chromatin structures on the two regions.

In addition of these major points, 3 reviewers gave comments shown below, some of which are redundant to the above points. It would be nice to respond (or rebut) them also.

*Reviewer #1 (Recommendations for the authors):*

1. Although the authors have not analyzed the defect of the eco1-aa mutant in meiotic recombination in detail, which would be for future studies, the authors need more detailed analysis of chromosome segregation at meiosis I and II by examining microtubule(spindle) dynamics to see the altered centromere and peri-centromeric regions affect the spindle dynamics to see a molecular role of the chromosomal regions in accurate segregation of chromosomes.

*Reviewer #2 (Recommendations for the authors):*

Below are a number of thoughts, critiques, and suggestions that might help to make the manuscript more impactful. They are arranged according to their order in the manuscript.

Line 76: Here and elsewhere (until the Discussion) there is de-emphasis of the fact that it is already known that Eco1 in yeast can act outside of S-phase to acetylate Smc3 (albeit at lower levels). This was shown in G1 and in cdc6 mutants in the two publications listed. I think it would be easier to state up front that this is known to occur.

Related – it might be worth mentioning that budding yeast Eco1 levels are down-regulated in vegetatively cells as they exit S-phase by (by CDK phosphorylation and degradation) – (Lyons and Morgan 2011)

Line 111: "In meiosis, Eco1 acetylates Smc3 independently of replication…." This result could appreciated in context if the known (G1, cdc6, and DNA damage) examples of Eco1 acetylating Smc3 in replication-independent scenarios had been spelled out.

Line111: Maybe should say "both linked to and independently of DNA replication". It's not all independent of replication.

Line 136: I'm not sure this is so "Surprising" if one is already aware that Eco1- family proteins are known to acetylate Smc3 outside of S-phase and in some situations much more than in S-phase. I personal feel the editorial starter words (Surprisingly, Interestingly, etc.) seem manipulative when I'm reading a Results section (although, yes, I do it myself sometimes – probably to be manipulative).

Line 163: "Therefore…." It's not ideal to have a tag that inactivates meiotic function even without the rapamycin added, but I understand the constraints of trying to modulate gene function in meiosis. So does this result mean that there is a meiosis-specific function (or at least one that is more essential in meiosis than mitosis) that involves the carboxy-end of Eco1?

Line 164: Do you mean – "cells that were induced"?

Line 165: Does "eco1-aa" stand for something?

Line 169: Does it have to be "later" stages? Not any time after cohesin establishment?

Line 184: "Therefore.." This sentence seems like it should be moved in front of the sentence that precedes it.

Line 194: What makes it remarkable?

Line 196: The cohesin defects of WPL1 mutations have been reported previously – though using a CEN5 marker (Challa et al.,?). Shouldn't this be mentioned here?

Line 197: Shouldn't it be "a" critical role? Is there only one?

Line 198: Maybe Eco1's essential function on arms is to counteract Wpl1, but only after Wpl1 does its other critical function -not cohesin unloading – the unappreciated function that is necessary for normal cohesion establishment/function. Wpl1 deletion shouldn't be expected to restore cohesion in eco1-aa since you have shown that wpl1 has a cohesion defect.

Line 198: Here and throughout the paper I thought this role of Wpl1 in promoting cohesion that you are revealing (even though Challa saw it previously) was under-discussed. It helps to show us that cohesion regulation isn't just a few genes each with a singular function (cohesion "off" or cohesion "on")

Figure 4: I found Figure S4 A and B to be very helpful. Could they be included in Figure 4?

Line 260: The "length" of contact stripe is a bit hard to know since it gets very dim. "Intensity – yes.

Line 265: Would it be more accurate to say the boundaries persisted in wpl1 (the ratio map w WT in Figure S4F looks mainly neutral at the borders), but the movement towards longer loop sizes in wpl1 makes the borders look brighter in Figure 4F because there is less red around the border stripes (ie. Better signal to noise, not a stronger signal)?

Line 290: The "interestingly" seems to suggest this observation is unprecedented. But there is evidence in budding yeast of low levels acetylation in unreplicated vegetative cells. Further, the demonstration earlier in the paper that in meiosis, Eco1 levels persist after S-phase into meiotic prophase (an important observation) would make this result an expected one, given that in other situations when the Eco family of transacteylases are present they will acetylate Smc3.

Line 295-339: I think these are important results (cohesin deposition in clb5 clb6 cells). But a critical control is to demonstrate that the observed clb5, clb6 cohesin patterns (Figure 5) are not residual cohesins from the preceding mitotic cycle and were indeed deposited during the replication-free meiosis. I don't think it is critical to the manuscript but a time course of cohesin loading in clb5, clb6 would be informative (T=0 to T=6).

Line 371: The apparent disjunction of sisters in the spo1 eco1-aa cells would, if true, be an important observation, making budding yeast a little more like other organisms with respect to co-orientation of sisters. However, an alternate explanation is that in the eco1-aa the sisters simply become totally disconnected before metaphase I and separate independently: 50% to opposite poles (pretty close to 40%) and 50% to the same pole. If in eco1-aa prophases, 20% of sisters stayed connected and co-oriented and 80% lost track of each other and segregated randomly the cells would exhibit the observed result. This could be resolved with a video. If the sisters orient mid-spindle as a single unit (one GFP spot) in prometaphase then disjoin at anaphase I, the reported conclusions are justified. If sisters behave as two independent GFP foci in pro-metaphase then partition randomly a different interpretation (that cohesin holds sisters together in prophase) is required.

Line 406: "Therefore…". I'm confused by this conclusion. The experiments show Eco1 is required at meiosis II for something that isn't just amounts of cohesin, Sgo1 or Spo13. And that "thing" is necessary for MII bi-orientation or cohesion – but it isn't clear what the thing is.

With respect to post-replicative Eco1 activity in meiosis. Is it worth discussing the persistence if Eco1 in meiotic prophase reported here relative Eco1 regulation in mitotic cycles? Is persistence related to the behaviors of Eco1 in cells with DNA damage since meiotic prophase cells are full of DSBs? Maybe not – just a thought.

*Reviewer #3 (Recommendations for the authors):*

1. The authors are encouraged to confirm some of the more distinctly unusual effects, such as the strong prophase arrest and the Wpl1-independent cohesin maintenance defect, using an independent eco1 allele, such as a ts allele or an auxin degron allele.

2. Figure 2D. Please provide separate graphs for spore formation and spore viability. This combined data is difficult to look at and also does not show whether there is a spore formation defect in the eco1 wpl1 double mutant.

3. Line 407: The conclusion "the failure to build rather than to protect" is an overstatement and should be softened. The authors show nicely that the mutants are not obviously defective in protection, but that the meiosis II missegregation is a defect in building cohesion has not been shown beyond correlation.

4. Figures 6B: Please provide better description of these data. Were these numbers taken at a particular time point or at the end of the live cell imaging? Also, please provide time course data as in 6A for the meiotic progression of experiments in 7E and S7A.

---

## [Author Response]

Essential revisions:1. HiC analysis at 0 h in clb5, clb6: The authors nicely showed that the clb5 clb6 double mutant (and CDC6mn), which is defective in meiotic DNA replication, shows Eco1-dependent acetylation of Smc3 (Figure 1E and S1E). This is clearly difference from mitotic cells, which show replication-dependent Smc3 acetylation. The key finding in the paper would be on replication-independent border formation (Figure 5). One concern is whether this is indeed promoted by Smc3 acetylation or comes from chromosome memory of the boundary from previous mitotic cells. To distinguish these possibilities, the authors need to analyze boundary profiles in the clb5 clb6 double mutant (and wild type) at 0 h and compared to those in prophase I (Figure 5).

In early G1 following mitosis, separase remains active and so all cohesin should be cleaved. Also, Scc1 (or Rec8 when considering subsequent entry into meiosis) is not expressed until after START (we examined this previously in mitosis (Fernius et al., 2013)) and this is apparent for meiosis from Figure 1C, E and I in our current manuscript. Given the absence of chromosomal cohesin in early G1, and our prior demonstration that the boundaries are strictly dependent on cohesin (Paldi et al., 2020) together with the demonstration from the Neale group that Rec8 is required for loops in meiosis (Schalbetter et al., 2019), it seems unlikely that a chromosome memory of the boundary could persist from mitotic cells into meiosis. Moreover, both the Neale and Koszul groups showed that in the G1 phase before meiosis loops are absent and established only throughout meiotic S phase and prophase I (Muller et al., 2018; Schalbetter et al., 2019).

Nevertheless, we carried out the suggested experiment, performing Hi-C on meiotic G1 arrest (*IME1, IME4* depletion), which is prior to S phase and prior to Rec8 expression. The Hi-C data for wild type and *clb5Δ clb6Δ* in the *ime1/ime4* pre-S phase block are shown together with the prophase I data (*ndt80Δ*) in Figure 6 and Figure 6 —figure supplement 1. The results show that in both wild type and *clb5Δ clb6Δ* cells anchored loops (or any kind of contact) are absent at meiotic entry. Therefore, the boundaries observed in prophase are established de novo in meiosis during S phase/prophase and, at least in part, independently from DNA replication.

To further address the question of whether the boundaries in unreplicated cells are due to Eco1-dependent acetylation we also performed Hi-C in unreplicated prophase I cells lacking Eco1 function. For reasons that are unclear, we found that *eco1-aa clb5Δ clb6Δ* cells are inviable, and so we performed Hi-C on prophase I-arrested *eco1-aa cdc6-md* cells along with *cdc6-md, eco1-aa* and wild type controls (Figure 7; Figure 7 —figure supplement 2). We also provide matched Smc3 ChIP-seq for these strains in prophase I (Figure 7—figure supplement 1). As in replicated cells (Figures 4 and 5), Eco1 is required for robust border/boundary formation in unreplicated (*cdc6-md*) cells.

Therefore we added three new major datasets: (i) wild type and *clb5Δ clb6Δ* Hi-C in meiotic G1; (ii) wild type, *cdc6-md, eco1-aa* and *cdc6-md eco1-aa* Hi-C in prophase I and (iii) wild type, *cdc6-md, eco1-aa* and *cdc6-md eco1-aa* ChIP-seq in prophase I. Together, these datasets show that (i) borders/boundaries are established de novo in meiosis during S phase/prophase; (ii) borders/boundaries and loops do not depend on DNA replication and (iii) Eco1 can drive loop formation and anchoring independently of DNA replication, presumably by acetylating loop extruding cohesin.

2. Live-imaging of sister centromeres during meiosis I: The role of Eco1-dependent Smc3 acetylation in promoting mono-polar kinetochore formation is one of the key findings (Figure 6) since previous studies showed mono-polar orientation of sister kinetochores during meiosis I does not depend on Rec8-cohesin in budding yeast. The authors only analyzed the results of chromosome segregation after anaphase I of the spo11 eco1-aa mutant. It would be helpful if the authors could see bipolar sister kinetochores on meiotic spindles during meiosis I. One concern is on precocious sister separation in late prophase I may cause similar defect in the assay (Figure 6C and D). The authors can check the status of sister centromeres in late prophase I, metaphase I and anaphase I in the spo11 eco1-aa mutant by live-imaging of cells with CenV-GFP used in Figures 2 and 6.

We apologise for the lack of clarity in our description of the live cell imaging experiment and its conclusions. Our data show that the mono-orientation defect is likely a *consequence* of the centromeric cohesion defect. We therefore believe that cohesion at centromeres in prophase I is required for the establishment of monoorientation. This is an important conclusion as it differs from the previous conclusion (Monje-Casas et al., 2007) which suggested that the monopolin complex linked sister centromeres independently of cohesin. Monje-Casas based their conclusion on the finding that sister centromere GFP foci are not separated in *spo11Δ rec8Δ* prophase I cells, but they are in *spo11Δ rec8Δ mam1Δ* double mutants (Figure 7 in Monje-Casas et al., 2007).

We now present data showing the status of *CEN5-*GFP cohesion in prophase I cells from the live cell imaging experiment (Figure 8C). Only cells that progress to anaphase I were included in this analysis to ensure that any effects observed were not a result of prolonged prophase I arrest. Analysis of metaphase I cells is confounded by the fact that in *spo11Δ* background, spindles elongate directly after spindle pole body duplication as there are no linkages between homologs. We therefore restricted our analysis to prophase I (new Figure 8C) and anaphase I (as was already shown in our original submission). As shown in Figure 8C, the only mutants to show splitting of *CEN5-GFP* foci prior to SPB separation are *spo11Δ eco1-aa* and *eco1-aa* (though very few *eco1-aa* cells progressed beyond prophase I). The absence of centromeric cohesion in *spo11Δ eco1-aa* cells led to equational segregation of sister chromatids in meiosis I. Notably, these defects are all rescued by *wpl1Δ.* This, together with the data presented in Figure 2F leads to the conclusion that centromeric cohesion is required for the establishment of sister kinetochore monoorientation. We have revised the text describing this experiment to indicate the likely causal relationship between centromeric cohesion and sister kinetochore monoorientation.

3. Discussion on functionality of Eco1-FRB-GFP: Since the FRB-GFP tag seems to affect the functionality of Eco1 protein in meiosis, it would be nice to carry out a complemental approach such as auxin-degron. However, there might be similar problem(s) for any tagging, even for N-terminal tagging. Thus, it is recommended that, by text rewriting, the authors describe more clearly about the limitation of the tag to know the Eco1 function in vivo and soften some conclusions in the text.

We do not consider that lack of functionality of the FRB-GFP tag in meiosis a major issue as our intention is to ablate Eco1 function in meiosis. We have not performed any experiment that relies on the functionality of the tag (such as depleting Eco1 function for a specific window – we agree that such experiments are impossible with this allele). Rather, *eco1-aa* can be considered similar to a meiotic depletion allele such as *pCLB2* where genes are placed under control of a mitosis-specific promoter and which are widely used in the literature (original description (Lee and Amon, 2003)). The only slight concern would be if *eco1-aa* has dominant effects, even in the presence of rapamycin (note that all our experiments except the initial characterisation were carried out in the presence of rapamycin from the time that we induce meiosis). However, dominant effects seem very unlikely, as evidence indicates that *eco1-aa* is a loss of function allele. First, Smc3-Ac is greatly diminished in *eco1-aa* (Figure 1I), similar to *eco1Δ wpl1Δ* vegetative cells. Second, deletion of *WPL1* rescues the viability of *eco1-aa* cells, similar to what has been shown for *eco1Δ* (Figure 1H).

4. Figure 7H and a model: In this model, the authors suggest the model on a cohesin-mediated loop-extrusion and sister chromatid cohesion. As, in the paper, the authors showed two distinct cohesin-mediated structures on centromeres and chromosome arms (and pericentromeres). It would be helpful to readers if the authors provide models on how cohesin- and its acetylation-mediated chromatin structures on the two regions.

Thank you for this suggestion, we have now revised the model to include the centromere and pericentromere borders and included this as a separate Figure 10.

In addition of these major points, 3 reviewers gave comments shown below, some of which are redundant to the above points. It would be nice to respond (or rebut) them also.Reviewer #1 (Recommendations for the authors):1. Although the authors have not analyzed the defect of the eco1-aa mutant in meiotic recombination in detail, which would be for future studies, the authors need more detailed analysis of chromosome segregation at meiosis I and II by examining microtubule(spindle) dynamics to see the altered centromere and peri-centromeric regions affect the spindle dynamics to see a molecular role of the chromosomal regions in accurate segregation of chromosomes.

Please see response to essential revisions #2 above. In particular, please note that due to the necessity to analyse chromosome segregation in *spo11Δ* background, where homologs are not linked, leading to premature extension of meiosis I spindles, it is unclear how informative examination of the spindle directly would be. We therefore chose to focus our analysis on *CEN5-GFP*, using the spindle pole bodies labels to stage the meiotic cells.

Reviewer #2 (Recommendations for the authors):Below are a number of thoughts, critiques, and suggestions that might help to make the manuscript more impactful. They are arranged according to their order in the manuscript.Line 76: Here and elsewhere (until the Discussion) there is de-emphasis of the fact that it is already known that Eco1 in yeast can act outside of S-phase to acetylate Smc3 (albeit at lower levels). This was shown in G1 and in cdc6 mutants in the two publications listed. I think it would be easier to state up front that this is known to occur.

In the two publications referred to, the situation is not so clear-cut as suggested by the reviewer. In one study (Beckouët et al., 2010), the authors delayed replication (by delaying Dbf4 expression) which delayed Smc3 acetylation. In the other study (Ben-Shahar et al., 2008), Cdc6 is depleted but residual replication occurs, which could explain the small amount of Smc3 acetylation observed. The best evidence that Eco1 can function independently from DNA replication comes from studies by the Morgan and Nasmyth groups who showed that Eco1 activity after S phase can establish cohesin de novo (Beckouët et al., 2010; Lyons and Morgan, 2011). In addition, it is known that Eco1 can establish cohesin in G2 in response to DNA damage (Heidinger-Pauli et al., 2009; Lyons et al., 2013). To take account of these observations, we modified the text referring to these experiments in the introduction.:

Related – it might be worth mentioning that budding yeast Eco1 levels are down-regulated in vegetatively cells as they exit S-phase by (by CDK phosphorylation and degradation) – (Lyons and Morgan 2011)

Please see comment above.

Line 111: "In meiosis, Eco1 acetylates Smc3 independently of replication…." This result could appreciated in context if the known (G1, cdc6, and DNA damage) examples of Eco1 acetylating Smc3 in replication-independent scenarios had been spelled out.

Please see comment above.

Line111: Maybe should say "both linked to and independently of DNA replication". It's not all independent of replication.

We made the suggested change, though the fraction of acetylated Smc3 is similar in replicated and unreplicated cells (Figure 1F) so the importance of replication is unclear.

Line 136: I'm not sure this is so "Surprising" if one is already aware that Eco1- family proteins are known to acetylate Smc3 outside of S-phase and in some situations much more than in S-phase. I personal feel the editorial starter words (Surprisingly, Interestingly, etc.) seem manipulative when I'm reading a Results section (although, yes, I do it myself sometimes – probably to be manipulative).

We re-worded this sentence to remove “surprisingly”.

Line 163: "Therefore…." It's not ideal to have a tag that inactivates meiotic function even without the rapamycin added, but I understand the constraints of trying to modulate gene function in meiosis. So does this result mean that there is a meiosis-specific function (or at least one that is more essential in meiosis than mitosis) that involves the carboxy-end of Eco1?

We do not know the reason but we suspect that C-terminal tags on Eco1 reduce its stability, particularly in meiosis. We observed a similar meiosis-specific loss of function with Eco1-6HA (not shown), though Eco1-6His-3FLAG is functional, as shown in Figure 1—figure supplement 1A. This would clearly be a problem if our aim was to re-activate Eco1 at a particular time in meiosis but as our goal is merely to inactivate Eco1 in meiosis we do not consider the lack of functionality of *eco1-aa* a major problem. It could be compared to e.g. temperature-sensitive alleles which have used in numerous studies and which are rarely fully functional at the permissive temperature. The problem of inactivating proteins in meiosis is an important consideration. In general, and not specific for Eco1, our attempts to induce degradation using the auxin-inducible degron have not produced satisfactory results (and this would also require the addition of a tag to Eco1). Temperature-sensitive alleles suffer from the difficulty that meiosis is itself temperature-sensitive.

Line 164: Do you mean – "cells that were induced"?

This has been corrected.

Line 165: Does "eco1-aa" stand for something?

“Anchor-away”, we added this in brackets to this sentence.

Line 169: Does it have to be "later" stages? Not any time after cohesin establishment?

We changed this sentence to:

“We reasoned that Eco1 activity early in meiosis may be required to counter Wpl1-dependent cohesin destabilization during any of the subsequent stages of meiosis.”

Line 184: "Therefore.." This sentence seems like it should be moved in front of the sentence that precedes it.

We changed this.

Line 194: What makes it remarkable?

In our opinion, the remarkable finding from these experiments is that there are locus-specific effects i.e. that *wpl1Δ* rescues the cohesion defect of *eco1-aa* at centromeres, but not on chromosome arms. We clarified this as follows:

“Remarkably, we observed distinct effects of *WPL1* deletion in *eco1-aa* cells on centromeres and chromosome arms: deletion of *WPL1* restored cohesion at the centromere (*CEN5-GFP*), but not at the chromosomal arm site (*LYS2-GFP*) (Figure 2F and H).”

Line 196: The cohesin defects of WPL1 mutations have been reported previously – though using a CEN5 marker (Challa et al.,?). Shouldn't this be mentioned here?

Thank you for pointing this out, we now cited this observation. Of note, the observations with the *CEN5* marker differ in our study and the Challa study, for reasons that are unclear. One possibility is the position of the *CEN5* labels in the two study could be different.

Line 197: Shouldn't it be "a" critical role? Is there only one?

We changed as suggested.

Line 198: Maybe Eco1's essential function on arms is to counteract Wpl1, but only after Wpl1 does its other critical function -not cohesin unloading – the unappreciated function that is necessary for normal cohesion establishment/function. Wpl1 deletion shouldn't be expected to restore cohesion in eco1-aa since you have shown that wpl1 has a cohesion defect.

*wpl1Δ* cells show almost wild type meiosis and spore viability (Figure 2, Figure 2—figure supplement 2, Challa et al., 2016). Chromosome segregation occurs with high fidelity in *wpl1Δ* cells (Figure 8). These observations argue against a significant cohesion defect in *wpl1Δ* cells. The only observation suggestive of cohesion defects in *wpl1Δ*cells are those made by Challa (2016) and ourselves (Figure 2F) where sister GFP labels are observed to separate in a prophase I arrest. An alternative explanation for this observation is that the longer chromatin loops in *wpl1Δ* cells lead to extrusion of part of the large (~10kb) *tetO* array away from its sister sequence. Since there is no compelling evidence for *wpl1Δ* cells having significant cohesion defects, the more likely explanation is that Eco1’s essential function on arms is to stabilize chromatin loops.

Line 198: Here and throughout the paper I thought this role of Wpl1 in promoting cohesion that you are revealing (even though Challa saw it previously) was under-discussed. It helps to show us that cohesion regulation isn't just a few genes each with a singular function (cohesion "off" or cohesion "on")

As argued in the point above, there is no compelling data arguing for a role of Wpl1 in cohesion (in fact there is compelling data suggesting that cohesion is basically functional). We now state this explicitly. “Importantly, *wpl1Δ* has only a minor effect on meiosis and spore viability (Figure 2—figure supplement 2; (Challa et al., 2016)). Furthermore, chromosome segregation fidelity in *wpl1Δ* cells is comparable to wild type (see below), indicating that cohesion is largely functional.” We believe that any further discussion of the *wpl1Δ* phenotype detracts from the main message which is that Eco1 is required to stabilize loops and cohesion.

Figure 4: I found Figure S4 A and B to be very helpful. Could they be included in Figure 4?

Thank you for the suggestion, we have now included these plots in the main figures but had to split Figure 4 into Figure 4 (chromosome organisation genome-wide) and Figure 5 (focus on the pericentromere) for space reasons. The ratio maps appear in Figure 5.

Line 260: The "length" of contact stripe is a bit hard to know since it gets very dim. "Intensity – yes.

We re-worded this to: “The intensity of the Hi-C contact stripe protruding from centromeres increased at progressively longer distances in *eco1-aa* and *wpl1Δ eco1-aa* cells, compared to wild type”

Line 265: Would it be more accurate to say the boundaries persisted in wpl1 (the ratio map w WT in Figure S4F looks mainly neutral at the borders), but the movement towards longer loop sizes in wpl1 makes the borders look brighter in Figure 4F because there is less red around the border stripes (ie. Better signal to noise, not a stronger signal)?

This is a good point, we re-worded this to read:

“Pile-ups centered on all 32 pericentromere borders (Paldi et al., 2020) revealed strong boundaries in wild type, which become sharper and more defined in *wpl1Δ* cells (Figure 4F; Figure S4F).”

Line 290: The "interestingly" seems to suggest this observation is unprecedented. But there is evidence in budding yeast of low levels acetylation in unreplicated vegetative cells. Further, the demonstration earlier in the paper that in meiosis, Eco1 levels persist after S-phase into meiotic prophase (an important observation) would make this result an expected one, given that in other situations when the Eco family of transacteylases are present they will acetylate Smc3.

We removed “interestingly”.

Line 295-339: I think these are important results (cohesin deposition in clb5 clb6 cells). But a critical control is to demonstrate that the observed clb5, clb6 cohesin patterns (Figure 5) are not residual cohesins from the preceding mitotic cycle and were indeed deposited during the replication-free meiosis. I don't think it is critical to the manuscript but a time course of cohesin loading in clb5, clb6 would be informative (T=0 to T=6).

We had actually not examined cohesin deposition in *clb5Δ clb6Δ* cells in the previous version of our manuscript, only Hi-C. We chose not to perform this experiment because cohesin is not expressed at t=0 and would therefore not be expected to associate with the chromosomes. This point is addressed in essential revisions #1.

Line 371: The apparent disjunction of sisters in the spo1 eco1-aa cells would, if true, be an important observation, making budding yeast a little more like other organisms with respect to co-orientation of sisters. However, an alternate explanation is that in the eco1-aa the sisters simply become totally disconnected before metaphase I and separate independently: 50% to opposite poles (pretty close to 40%) and 50% to the same pole. If in eco1-aa prophases, 20% of sisters stayed connected and co-oriented and 80% lost track of each other and segregated randomly the cells would exhibit the observed result. This could be resolved with a video. If the sisters orient mid-spindle as a single unit (one GFP spot) in prometaphase then disjoin at anaphase I, the reported conclusions are justified. If sisters behave as two independent GFP foci in pro-metaphase then partition randomly a different interpretation (that cohesin holds sisters together in prophase) is required.

The data presented in Figure 6 of the original manuscript was all from videos. As argued in our response to essential revisions #2, we believe that our findings show that Eco1 is required for centromeric cohesion, which is in turn required for sister kinetochore monoorientation, in contrast to an earlier conclusion.

Line 406: "Therefore…". I'm confused by this conclusion. The experiments show Eco1 is required at meiosis II for something that isn't just amounts of cohesin, Sgo1 or Spo13. And that "thing" is necessary for MII bi-orientation or cohesion – but it isn't clear what the thing is.

We toned down this conclusion. It now reads “Therefore, a failure in cohesin protection is unlikely to be the cause of meiosis II mis-segregation in *eco1-aa* cells. Since *wpl1Δ* rescues the loss of cohesin on pericentromeric borders in *eco1-aa* cells, but not the anchoring of loops, it is likely that Eco1 enables accurate meiosis II segregation by anchoring cohesin at chromatin boundaries to generate robust cohesion at pericentromere borders.”

With respect to post-replicative Eco1 activity in meiosis. Is it worth discussing the persistence if Eco1 in meiotic prophase reported here relative Eco1 regulation in mitotic cycles? Is persistence related to the behaviors of Eco1 in cells with DNA damage since meiotic prophase cells are full of DSBs? Maybe not – just a thought.

We apologise for the confusion on the timing of Eco1 inactivation due to our mislabelling of meiotic stages on the western blot. As pointed out by Reviewer 3, FACS profile from the timecourse in Figure 1A-C shows that replication was complete by timepoint 150min, not 120min as we indicated on top of the Western blots and described in the text. The data show that Eco1 degradation starts soon after the completion of DNA replication. We have adjusted Figure 1C and the text accordingly. Nonetheless, low levels of Eco1 reproducibly persist all throughout meiosis, opening up the possibility that Eco1 could acetylate Smc3, or other targets, after S phase. Although Eco1-dependent acetylation of Smc3 occurs in *spo11Δ* cells, and therefore in the absence of meiotic double strand breaks (Figure 1 —figure supplement 2), it is possible that Eco1 activity is boosted by DNA damage in meiosis similar to in mitosis. We agree that this is a very interesting avenue for future investigation, but as the manuscript already covers a lot of ground we decided to focus our discussion on the main findings.

Reviewer #3 (Recommendations for the authors):1. The authors are encouraged to confirm some of the more distinctly unusual effects, such as the strong prophase arrest and the Wpl1-independent cohesin maintenance defect, using an independent eco1 allele, such as a ts allele or an auxin degron allele.

Unfortunately, it is challenging to work with ts alleles in meiotic cells as meiosis is itself temperature-sensitive. Furthermore, ts alleles are never fully functional at the permissive temperature so effects due to impaired function in the mitotic divisions preceding meiosis can also not be ruled out, so even if they were an option, it is unlikely that they would provide any advantage over *eco1-aa*. Despite attempts to make several aid degrons, we have not been successful with reliable depletion in meiosis. Furthermore, it is likely that aid tagging will have the same effect on Eco1 protein function, since we know that other tags on Eco1 are non-functional. We therefore believe that *eco1-aa* represents the best possible available tool to analyse Eco1 function in meiosis.

2. Figure 2D. Please provide separate graphs for spore formation and spore viability. This combined data is difficult to look at and also does not show whether there is a spore formation defect in the eco1 wpl1 double mutant.

We agree that this graph is confusing – in fact it only shows spore viability where the aim was to find 50 tetrads to dissect. It was only possible to find 3 tetrads for the *eco1-aa* cells. We have therefore simply removed the data for *eco1-aa* from the graph as we agree that it is not informative. Spore formation for all strains is shown in Figure 2—figure supplement 2A.

3. Line 407: The conclusion "the failure to build rather than to protect" is an overstatement and should be softened. The authors show nicely that the mutants are not obviously defective in protection, but that the meiosis II missegregation is a defect in building cohesion has not been shown beyond correlation.

We agree. As described in our response to reviewer 2, we have softened this conclusion to read: ““Therefore, a failure in cohesin protection is unlikely to be the cause of meiosis II mis-segregation in *eco1-aa* cells. Since *wpl1Δ* rescues the loss of cohesin on pericentromeric borders in *eco1-aa* cells, but not the anchoring of loops, it is likely that Eco1 enables accurate meiosis II segregation by anchoring cohesin at chromatin boundaries to generate robust cohesion at pericentromere borders.”

4. Figures 6B: Please provide better description of these data. Were these numbers taken at a particular time point or at the end of the live cell imaging? Also, please provide time course data as in 6A for the meiotic progression of experiments in 7E and S7A.

Every cell that had 1 SPB at the start of the video was scored for the duration of the video. The percentage of cells that displayed 2 SPBs or 4 SPBs for each strain is shown. A small number of mitotic/dead cells (<1% for each strain) were excluded from the analysis. We have now included this information in the figure legend. As we found the live imaging to give reliable information comparable to the fixed cell data, we decided not to perform the time course for the strains in 7E and S7A (Now Figure 9E-G and Figure 9—figure supplement 3).